# RAPL: A Relation-Aware Prototype Learning Approach for Few-Shot Document-Level Relation Extraction

**Shiao Meng, Xuming Hu, Aiwei Liu, Shu'ang Li, Fukun Ma, Yawen Yang, Lijie Wen**[*]

School of Software, Tsinghua University

{msa21,hxm19,liuaw20,lisa18,mfk22,yyw19}@mails.tsinghua.edu.cn

wenlj@tsinghua.edu.cn

## Abstract

How to identify semantic relations among entities in a document when only a few labeled documents are available? Few-shot document-level relation extraction (FSDLRE) is crucial for addressing the pervasive data scarcity problem in real-world scenarios. Metric-based meta-learning is an effective framework widely adopted for FSDLRE, which constructs class prototypes for classification. However, existing works often struggle to obtain class prototypes with accurate relational semantics: 1) To build prototype for a target relation type, they aggregate the representations of all entity pairs holding that relation, while these entity pairs may also hold other relations, thus disturbing the prototype. 2) They use a set of generic NOTA (*none-of-the-above*) prototypes across all tasks, neglecting that the NOTA semantics differs in tasks with different target relation types. In this paper, we propose a relation-aware prototype learning method for FSDLRE to strengthen the relational semantics of prototype representations. By judiciously leveraging the relation descriptions and realistic NOTA instances as guidance, our method effectively refines the relation prototypes and generates task-specific NOTA prototypes. Extensive experiments demonstrate that our method outperforms state-of-the-art approaches by average 2.61% $F_1$ across various settings of two FSDLRE benchmarks.[1]

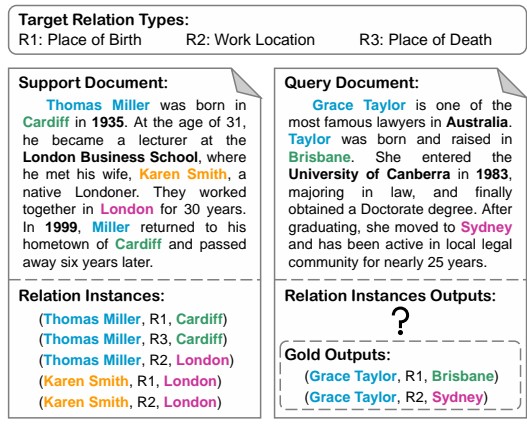

Figure 1: Illustration of a 1-Doc FSDLRE task. Entity mentions involved in relation instances are colored and in bold. Other mentions are also in bold for clarity.

## 1 Introduction

Document-level relation extraction (DocRE) aims to identify the relations between each pair of entities within a document, which is crucial for extracting complex cross-sentence relations and implementing large-scale information extraction (Zhou et al., 2021; Xie et al., 2022; Wei and Li, 2022; Sun et al., 2023). However, the annotation of DocRE data is both time-consuming and labor-intensive,

and many specific domains often lack annotated documents, making data scarcity a common issue in real-world scenarios. This motivates us to explore few-shot document-level relation extraction (FSDLRE) (Popovic and Färber, 2022). We illustrate an example of the FSDLRE task under the 1-Doc setting in Figure 1, where only one annotated support document is given along with three target relation types: Place of Birth, Work Location and Place of Death. The task is to predict all instances of the target relation types for pre-given entities in the query document, such as (Grace Taylor, Place of Birth, Brisbane).

Current efforts on FSDLRE (Popovic and Färber, 2022) mainly adopt the popular metric-based meta-learning framework (Vinyals et al., 2016; Snell et al., 2017), which aims to learn a metric space in which classification can be performed by computing distances to prototype representations of each class. By training on a collection of sampled FSDLRE tasks, the model learns general knowledge for FSDLRE, enabling rapid generalization to new tasks with novel relation types.

Ideally, within the metric-based paradigm, prototype representations should accurately capture

---

[*]Corresponding author.

[1]The data and code are available at https://github.com/THU-BPM/RAPL.

the relational semantics of each category. However, this can be challenging for existing FSDLRE methods: (1) Considering that an entity pair may express multiple relations in a document, if a relation prototype is obtained by aggregating the representations of entity pairs in the support set holding that relation, the relational semantics of the prototype is inevitably disturbed by irrelevant relations, thus affecting the discriminability of the metric space, as depicted in Figure 2(a). (2) Since most query entity pairs do not express any target relation, NOTA (*none-of-the-above*) is also considered as a category. Given that the target relation types vary for different tasks, if we merely introduce a set of learnable vectors as NOTA prototypes and apply them to all tasks, this "one-size-fits-all" strategy could result in the NOTA prototypes deviating from ideal NOTA semantics in certain tasks, thereby confusing the classification. As shown in Figure 2(a), the set of generic NOTA prototypes seems reasonable for task 1, while does not work well for task 2.

To address the two aforementioned issues in FSDLRE, we propose a novel **R**elation-**A**ware **P**rototype **L**earning method (RAPL). First, for each entity pair that holds relations in the support document, we leverage the inherent relational semantics in relation descriptions as guidance, deriving an instance-level representation for each expressed relation, as illustrated in Figure 2(b). The relation prototype is constructed by aggregating the representations of all its support relation instances, thus better focusing on relation-relevant information. Based on the instance-level support embeddings, we propose a relation-weighted contrastive learning method to further refine the prototypes. By incorporating inter-relation similarities into a contrastive objective, we can better distinguish the prototypes of semantically-close relations. Moreover, we design a task-specific NOTA prototype generation strategy. For each task, we adaptively select support NOTA instances and fuse them into a set of learnable base NOTA prototypes to generate task-specific NOTA prototypes, which more effectively capture the NOTA semantics in each task.

In summary, our main contributions are as follows: (1) We propose a novel relation-aware prototype learning method (RAPL) for FSDLRE, which effectively enhances the relational semantics of prototype representations. (2) In RAPL, we reframe the construction of relation prototypes into instance level and further propose a relation-weighted con-

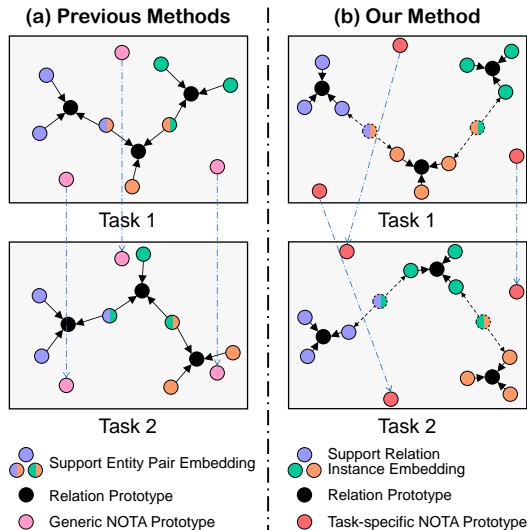

Figure 2: Embedding space illustration of previous methods (left) and our method (right). Task 1&2 are two FSDLRE tasks with different target relation types.

trastive learning method to jointly refine the relation prototypes. We also design a task-specific NOTA prototype generation strategy to better capture the NOTA semantics in each task. (3) Experiments demonstrate that our method outperforms state-of-the-art baselines by average 2.61% in $F_1$ across various settings of two FSDLRE benchmarks.

## 2 Problem Formulation

Few-shot document-level relation extraction is defined with an $N$-Doc setting (Popovic and Färber, 2022). In each individual FSDLRE task (also called an episode[2]), there are a set of $N$ support documents $\{D_{S,1}, ..., D_{S,N}\}$ and a query document $D_Q$, and the entity mentions in each document are pre-annotated. For each support document $D_{S,i}$, there is also a triple set $\mathcal{T}_{S,i}$ containing all valid $(e_h, r, e_t)$ triples in the document. Here $e_h$ and $e_t$ are the head and tail entity of a relation instance, and $r \in \mathcal{R}_{episode}$ is a relation type, with $\mathcal{R}_{episode}$ being the relation type set for which instances are to be extracted. The annotations of support documents are complete, which means any entity pair for which no relation type has been assigned can be considered as NOTA. Given these as inputs, the FSDLRE task aims to predict the triple set $\mathcal{T}_Q$ for the query document $D_Q$, which contains all valid triples in $D_Q$ of relation types in $\mathcal{R}_{episode}$.

Our approach follows the typical meta-learning

---

[2]In this paper, we use the terms "task" and "episode" interchangeably, which refer to the same concept.

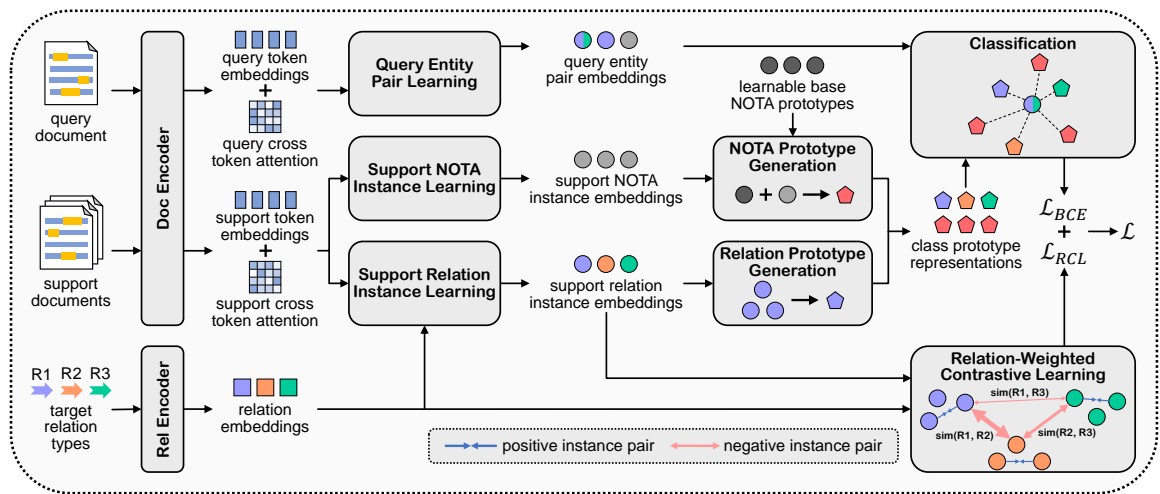

Figure 3: The overall architecture of our proposed RAPL method.

paradigm. In training phase, we construct a group of training episodes by sampling support and query documents from a training document corpus $\mathcal{C}_{train}$. The set $\mathcal{R}_{episode}$ of each training episode is a subset of $\mathcal{R}_{train}$, a relation type set for meta-training. The model aims to learn general knowledge from these training tasks to better generalize to novel tasks. In test phase, the model is evaluated on a group of test episodes sampled from a test document corpus $\mathcal{C}_{test}$, which is disjoint with $\mathcal{C}_{train}$. The set $\mathcal{R}_{episode}$ of each test episode is a subset of $\mathcal{R}_{test}$, a relation type set for meta-testing, which is also disjoint with $\mathcal{R}_{train}$.

## 3 Methodology

The overall architecture of RAPL is illustrated in Figure 3. We first introduce the encoding procedure for documents and entities in Section 3.1. In Section 3.2 and Section 3.3, we elaborate on the learning of relation-aware relation prototypes and NOTA prototypes respectively. The training and inference processes are finally given in Section 3.4.

### 3.1 Document and Entity Encoding

We employ the pre-trained language model (Devlin et al., 2019) as the document encoder to encode each support or query document in a given episode. For each document $D$, we first insert a special token "*" at the start and end of each entity mention to mark the position of entity mentions. Then we feed the document into the encoder to obtain the contextualized token embeddings $\boldsymbol{H}$ and cross token attention $\boldsymbol{A}$:

$$\boldsymbol{H}, \boldsymbol{A} = \text{DocEncoder}(D), \qquad (1)$$

where $\boldsymbol{H} = [\boldsymbol{h}_1, \ldots, \boldsymbol{h}_{N_t}] \in \mathbb{R}^{N_t \times d}$, $N_t$ is the number of tokens in $D$, $d$ is the output dimension of encoder, and $\boldsymbol{A} = [\boldsymbol{a}_1, \ldots, \boldsymbol{a}_{N_t}] \in \mathbb{R}^{N_t \times N_t}$ is the average of attention heads in the last encoder layer. We take the embedding of "*" before each entity mention as the corresponding mention embedding. For an entity $e_i$ mentioned $N_{e_i}$ times in the document via $\mathcal{M}_{e_i} = \{m_j^i\}_{j=1}^{N_{e_i}}$, we apply log-sumexp pooling (Jia et al., 2019) over mention embeddings to obtain the entity embedding $\boldsymbol{h}_{e_i} \in \mathbb{R}^d$: $\boldsymbol{h}_{e_i} = \log \sum_{j=1}^{N_{e_i}} \exp(\boldsymbol{h}_{m_j^i})$, where $\boldsymbol{h}_{m_j^i} \in \mathbb{R}^d$ is the embedding of $e_i$'s $j$-th mention.

### 3.2 Relation-Aware Relation Prototype Learning

For each target relation type in a given episode, we aim to obtain a prototype representation that can better capture the corresponding relational semantics. To this end, we first propose to construct the relation prototypes based on instance-level support embeddings, enabling each prototype to focus more on relation-relevant information in support documents. Then we propose an instance-level relation-weighted contrastive learning method, which further refines the relation prototypes.

#### 3.2.1 Instance-Based Prototype Construction

Given a relation instance $(e_h, r, e_t)$ in a support document, we first compute a pair-level importance distribution $\boldsymbol{a}^{(h,t)} \in \mathbb{R}^{N_t}$ over all tokens in the document to capture the context relevant to the entity pair $(e_h, e_t)$ (Zhou et al., 2021):

$$\boldsymbol{a}^{(h,t)} = \frac{\boldsymbol{a}_{e_h} \odot \boldsymbol{a}_{e_t}}{\boldsymbol{a}_{e_h}^{\mathsf{T}} \boldsymbol{a}_{e_t}}, \qquad (2)$$

where $\boldsymbol{a}_{e_h} \in \mathbb{R}^{N_t}$ is an entity-level attention obtained by averaging the mention-level attention $\boldsymbol{a}_{m_i^h} \in \mathbb{R}^{N_t}$ at the token "$*$" before $e_h$'s each mention $m_i^h$: $\boldsymbol{a}_{e_h} = \frac{1}{N_{e_h}} \sum_{i=1}^{N_{e_h}} \boldsymbol{a}_{m_i^h}$, likewise for $\boldsymbol{a}_{e_t}$, and $\odot$ is the Hadamard product. Meanwhile, we compute a relation-level attention distribution $\boldsymbol{a}^r \in \mathbb{R}^{N_t}$ over all tokens to capture the context relevant to relation $r$. We employ another pre-trained language model as the relation encoder, and concatenate the name and description of relation $r$ into a sequence, then feed the sequence into the encoder. We take the output embedding of "[CLS]" token as the relation embedding $\boldsymbol{h}_r \in \mathbb{R}^d$:

$$\boldsymbol{h}_r = \text{RelEncoder}(r), \qquad (3)$$

and compute the relation-level attention $\boldsymbol{a}^r$ as:

$$\boldsymbol{a}^r = \text{softmax}(\frac{\boldsymbol{H}\boldsymbol{W}\boldsymbol{h}_r}{\sqrt{d}}), \qquad (4)$$

where $\boldsymbol{W} \in \mathbb{R}^{d \times d}$ is a learnable parameter.

Based on $\boldsymbol{a}^{(h,t)}$ and $\boldsymbol{a}^r$, we further compute an instance-level attention distribution $\boldsymbol{a}^{(h,r,t)} \in \mathbb{R}^{N_t}$ over all tokens to capture the context relevant to the instance. Specifically, the value $a_i^{(h,r,t)}$ at $i$-th dimension of $\boldsymbol{a}^{(h,r,t)}$ is obtained by:

$$a_i^{(h,r,t)} = a_i^{(h,t)} + \mathbb{I}\big(i \in \text{top-}k\%(\boldsymbol{a}^{(h,t)} \odot \boldsymbol{a}^r)\big) \cdot a_i^r, \qquad (5)$$

where top-$k\%(\boldsymbol{x})$ returns the indices of the largest $k\%$ elements of $\boldsymbol{x}$, $k$ is a hyperparameter, and $\mathbb{I}$ is the indicator function. We also normalize $\boldsymbol{a}^{(h,r,t)}$ to regain the attention distribution. Here we do not use $\boldsymbol{a}^{(h,t)} \odot \boldsymbol{a}^r$ as the instance-level attention because, for an instance, the relation is expressed based on the entity pair. Multiplying them directly may erroneously increase the weight of tokens unrelated to the entity pair. Instead, we leverage relation-level attention to amplify the pair-level weight of the most relevant context with the instance.

Then, we compute an instance context embedding $\boldsymbol{c}^{(h,r,t)} \in \mathbb{R}^d$ by:

$$\boldsymbol{c}^{(h,r,t)} = \boldsymbol{H}^{\mathsf{T}} \boldsymbol{a}^{(h,r,t)}, \qquad (6)$$

and fuse it into the embeddings of head entity and tail entity to obtain the instance-aware entity representations $\boldsymbol{z}_h^{(h,r,t)}, \boldsymbol{z}_t^{(h,r,t)} \in \mathbb{R}^d$:

$$\boldsymbol{z}_h^{(h,r,t)} = \tanh(\boldsymbol{W}_h[\boldsymbol{h}_{e_h}; \boldsymbol{c}^{(h,r,t)}] + \boldsymbol{b}_h), \quad (7)$$

$$\boldsymbol{z}_t^{(h,r,t)} = \tanh(\boldsymbol{W}_t[\boldsymbol{h}_{e_t}; \boldsymbol{c}^{(h,r,t)}] + \boldsymbol{b}_t), \quad (8)$$

where $\boldsymbol{W}_h, \boldsymbol{W}_t \in \mathbb{R}^{d \times 2d}$, $\boldsymbol{b}_h, \boldsymbol{b}_t \in \mathbb{R}^d$ are learnable parameters. The instance representation of $(e_h, r, e_t)$ is then obtained by concatenating the head and tail entity representations, which we denote as $\boldsymbol{s}^{(h,r,t)} = [\boldsymbol{z}_h^{(h,r,t)}; \boldsymbol{z}_t^{(h,r,t)}] \in \mathbb{R}^{2d}$.

Finally, denoting the set of all instances of relation $r$ in support documents as $\mathcal{S}_r$, we compute the relation prototype $\boldsymbol{p}^r \in \mathbb{R}^{2d}$ by averaging the representations of relation instances in $\mathcal{S}_r$:

$$\boldsymbol{p}^r = \frac{1}{|\mathcal{S}_r|} \sum_{(e_h, r, e_t) \in \mathcal{S}_r} \boldsymbol{s}^{(h,r,t)}. \qquad (9)$$

### 3.2.2 Contrastive-Based Prototype Refining

By reframing the construction of relation prototypes into instance level, each prototype can better focus on relation-relevant support information. However, due to the complexity of document context, different instances of the same relation may exhibit varying patterns in expressing the relation, making it difficult for prototypes to capture the common relational semantics. Additionally, limited support instances make it challenging for prototypes of semantically-close relations to capture their deeper semantic differences. Therefore, we propose a relation-weighted contrastive learning method to further refine the relation prototypes.

Specifically, given an episode, we denote the set of all relation instances in support documents as $\mathcal{S}$, i.e., $\mathcal{S} = \bigcup_{r \in \mathcal{R}_{episode}} \mathcal{S}_r$. Also, for a relation instance $(e_h, r, e_t)$, we define the set $\mathcal{P}_{h,r,t} = \mathcal{S}_r \setminus \{(e_h, r, e_t)\}$ which contains all other instances in the support set that also express the relation $r$, and the set $\mathcal{A}_{h,r,t} = \mathcal{S} \setminus \{(e_h, r, e_t)\}$ which simply contains all other instances in the support set. Then we incorporate inter-relation similarities into a contrastive objective and define the relation-weighted contrastive loss $\mathcal{L}_{RCL}$ as:

$$\mathcal{L}_{RCL} = \frac{1}{|\mathcal{S}|} \sum_{(e_h, r, e_t) \in \mathcal{S}} \frac{-1}{|\mathcal{P}_{h,r,t}|} \sum_{(e_{\bar{h}}, r, e_{\bar{t}}) \in \mathcal{P}_{h,r,t}}$$

$$\frac{\exp(\boldsymbol{s}^{(h,r,t)} \cdot \boldsymbol{s}^{(\bar{h},r,\bar{t})}/\tau)}{\sum\limits_{(e_{\hat{h}}, \hat{r}, e_{\hat{t}}) \in \mathcal{A}_{h,r,t}} \omega_{r,\hat{r}} \cdot \exp(\boldsymbol{s}^{(h,r,t)} \cdot \boldsymbol{s}^{(\hat{h},\hat{r},\hat{t})}/\tau)},$$

$$(10)$$

$$\omega_{r,\hat{r}} = 1 + \mathbb{I}(r \neq \hat{r}) \cdot \frac{\text{cossim}(\boldsymbol{h}_r, \boldsymbol{h}_{\hat{r}}) + 1}{2}, \quad (11)$$

where $\tau$ is a hyperparameter and cossim denotes the cosine similarity. We argue that the proposal of this contrastive loss is non-trivial, considering

two aspects. First, it is difficult for previous methods to integrate with contrastive objective as they only obtain the pair-level support embeddings. The multi-label nature of entity pairs makes it difficult to define positive and negative pairs reasonably. Moreover, by incorporating inter-relation similarities, the proposed contrastive loss focuses more on pushing apart the instance embeddings of semantically-close relations, thus helping to better distinguish the corresponding relation prototypes.

### 3.3 Relation-Aware NOTA Prototype Learning

Since most query entity pairs do not hold any target relation, NOTA is also treated as a class. Existing methods typically learn a set of generic NOTA prototypes that are applied to all tasks, which may not be optimal in certain tasks since the NOTA semantics differs in tasks with different target relation types. To this end, we propose a task-specific NOTA prototype generation strategy to better capture the NOTA semantics in each individual task.

Concretely, we first introduce a set of learnable vectors $\mathcal{N}_{base} = \{\boldsymbol{p}_i^{base} \in \mathbb{R}^{2d}\}_{i=1}^{N_{nota}}$, where $N_{nota}$ is a hyperparameter. Unlike previous works that directly treat this set of vectors as NOTA prototypes, we regard them as base NOTA prototypes that need further rectification in each task. Since the annotation of support documents are complete, we have access to a support NOTA distribution which implicitly expresses the NOTA semantics. Therefore we resort to support NOTA instances to capture the NOTA semantics in each specific task. For a support NOTA instance $(e_h, nota, e_t)$, we use Equation 2 as the instance-level attention and obtain the instance representation $\boldsymbol{s}^{(h,nota,t)} = [\boldsymbol{z}_h^{(h,nota,t)}; \boldsymbol{z}_t^{(h,nota,t)}] \in \mathbb{R}^{2d}$ based on Equation 6~8. Denoting the set of all support NOTA instances as $\mathcal{S}_{nota}$, we adaptively select a NOTA instance for each base NOTA prototype $\boldsymbol{p}_i^{base}$:

$$
\begin{aligned}
(e_h, nota, e_t) = \underset{(e_h, nota, e_t) \in \mathcal{S}_{nota}}{\operatorname{argmax}} (\boldsymbol{s}^{(h,nota,t)} \cdot \boldsymbol{p}_i^{base} \\
- \max_{r \in \mathcal{R}_{episode}} \boldsymbol{s}^{(h,nota,t)} \cdot \boldsymbol{p}^r),
\end{aligned}
\tag{12}
$$

which locates the NOTA instance that is close to the base NOTA prototype while being far away from relation prototypes. Then we fuse it into $\boldsymbol{p}_i^{base}$ to obtain the final NOTA prototype $\boldsymbol{p}_i^{nota} \in \mathbb{R}^{2d}$:

$$
\boldsymbol{p}_i^{nota} = \alpha \boldsymbol{p}_i^{base} + (1 - \alpha) \boldsymbol{s}^{(h,nota,t)}, \tag{13}
$$

| Benchmark | Task | $N$ | $K$ (micro) | $K$ (macro) |
|---|---|---|---|---|
| FREDo | In-Domain 1-Doc | 2.18 | 2.36 | 2.24 |
| | In-Domain 3-Doc | 3.47 | 4.30 | 4.31 |
| ReFREDo | In-Domain 1-Doc | 3.50 | 3.50 | 3.11 |
| | In-Domain 3-Doc | 5.67 | 6.50 | 5.73 |
| FREDo & ReFREDo | Cross-Domain 1-Doc | 4.26 | 2.73 | 2.40 |
| | Cross-Domain 3-Doc | 6.08 | 5.55 | 5.27 |

Table 1: Average values for $N$ (way) and $K$ (shot) across test episodes in two benchmarks. $K$ (micro) denotes the average across all episodes, $K$ (macro) denotes the average of mean $K$ for each relation type.

where $\alpha$ is a hyperparameter. In this way, we obtain a set of task-specific NOTA prototypes which not only contain the general knowledge from meta-learning but also implicitly capture the NOTA semantics in each specific task.

### 3.4 Training Objective

Given an entity pair $(e_h, e_t)$ in the query document, we use Equation 2 as the pair-level attention and adopt a similar approach as Equation 6~8 to obtain the pair representation $\boldsymbol{q}^{(h,t)} = [\boldsymbol{z}_h^{(h,t)}; \boldsymbol{z}_t^{(h,t)}] \in \mathbb{R}^{2d}$. For each target relation type $r$ in the episode, we compute the probability of $r$ as:

$$
P_r^{(h,t)} = \frac{\exp(\boldsymbol{q}^{(h,t)} \cdot \boldsymbol{p}^r)}{\exp(\boldsymbol{q}^{(h,t)} \cdot \boldsymbol{p}^r) + \max_{i \in \mathcal{I}} \exp(\boldsymbol{q}^{(h,t)} \cdot \boldsymbol{p}_i^{nota})},
\tag{14}
$$

where $\mathcal{I} = \{1, ..., N_{nota}\}$. Then, denoting the set of all entity pairs in the query document as $\mathcal{Q}$, we compute the classification loss as:

$$
\begin{aligned}
\mathcal{L}_{BCE} = \frac{1}{|\mathcal{Q}|} \sum_{(e_h, e_t) \in \mathcal{Q}} - \sum_{r \in \mathcal{R}_{episode}} \\
(y_r^{(h,t)} \log(P_r^{(h,t)}) + (1 - y_r^{(h,t)}) \log(1 - P_r^{(h,t)})),
\end{aligned}
\tag{15}
$$

where $y_r^{(h,t)} = 1$ if $r$ exists between $(e_h, e_t)$, otherwise $y_r^{(h,t)} = 0$. The overall loss is defined as:

$$
\mathcal{L} = \mathcal{L}_{BCE} + \lambda \mathcal{L}_{RCL}, \tag{16}
$$

where $\lambda$ is a hyperparameter. During inference, we extract the relation instance $(e_h, r, e_t)$ in the query document if $\boldsymbol{q}^{(h,t)} \cdot \boldsymbol{p}^r > \max_{i \in \mathcal{I}} \boldsymbol{q}^{(h,t)} \cdot \boldsymbol{p}_i^{nota}$.

## 4 Experiments

### 4.1 Benchmarks and Evaluation Metric

We conduct experiments on the public FSDLRE benchmark FREDo (Popovic and Färber, 2022), and also construct ReFREDo, a revised version

| Model | FREDo | | | | ReFREDo | | | |
|---|---|---|---|---|---|---|---|---|
| | In-Domain | | Cross-Domain | | In-Domain | | Cross-Domain | |
| | 1-Doc $F_1$ | 3-Doc $F_1$ | 1-Doc $F_1$ | 3-Doc $F_1$ | 1-Doc $F_1$ | 3-Doc $F_1$ | 1-Doc $F_1$ | 3-Doc $F_1$ |
| DL-Base | 0.60 | 0.89 | 1.76 | 1.98 | 1.38 | 1.84 | 1.76 | 1.98 |
| DL-MNAV | $7.05 \pm 0.18$ | $8.42 \pm 0.64$ | $0.84 \pm 0.16$ | $0.48 \pm 0.21$ | $12.97 \pm 0.88$ | $12.43 \pm 0.36$ | $1.12 \pm 0.38$ | $2.28 \pm 0.19$ |
| DL-MNAV$_{SIE}$ | $7.06 \pm 0.15$ | $6.77 \pm 0.21$ | $1.77 \pm 0.60$ | $2.51 \pm 0.66$ | $13.37 \pm 0.98$ | $12.00 \pm 0.80$ | $1.39 \pm 0.74$ | $2.92 \pm 0.41$ |
| DL-MNAV$_{SIE+SBN}$ | $1.71 \pm 0.04$ | $2.79 \pm 0.24$ | $2.85 \pm 0.12$ | $3.72 \pm 0.14$ | $4.59 \pm 0.30$ | $5.43 \pm 0.24$ | $2.84 \pm 0.24$ | $3.86 \pm 0.27$ |
| KDDocRE | $2.59 \pm 0.71$ | $4.66 \pm 0.83$ | $1.03 \pm 0.31$ | $2.00 \pm 0.46$ | $4.76 \pm 0.55$ | $9.02 \pm 0.64$ | $2.30 \pm 0.59$ | $3.61 \pm 0.43$ |
| RAPL (Ours) | $\mathbf{8.75 \pm 0.80}$ | $\mathbf{10.67 \pm 0.77}$ | $\mathbf{3.33 \pm 0.50}$ | $\mathbf{5.35 \pm 0.72}$ | $\mathbf{15.20 \pm 0.82}$ | $\mathbf{16.35 \pm 0.60}$ | $\mathbf{3.51 \pm 0.79}$ | $\mathbf{5.48 \pm 0.63}$ |

Table 2: Results on FREDo and ReFREDo benchmarks. Reported results are macro averages across relation types. The best and second best performance methods are denoted in bold and underlined respectively.

of FREDo which resolves the annotation errors, enabling more reliable evaluation.

**FREDo** consists of two main tasks, an in-domain and a cross-domain task. For in-domain tasks, the training and test document corpus are from the same domain. For cross-domain tasks, the test documents are taken from a different domain, leading to wider disparities of text style, document topic and relation types between training and test. Each task has a 1-Doc and a 3-Doc subtask to measure the scalability of models. FREDo uses the training set of DocRED (Yao et al., 2019) as the training and development document corpus, the development set of DocRED as the in-domain test document corpus, and the whole set of SciERC (Luan et al., 2018) as the cross-domain test document corpus. The relation type set of DocRED is split into 3 disjoint sets for training (62), development (16) and in-domain test (18) in FREDo. FREDo samples 15k episodes for in-domain evaluation and 3k episodes for cross-domain evaluation.

Considering that FREDo uses DocRED, which suffers from the problem of incomplete annotation (Huang et al., 2022; Tan et al., 2022b), as the underlying document corpus, the episodes constructed in FREDo may also inherit these annotation errors. Therefore, we construct **ReFREDo** as a revised version of FREDo. In ReFREDo, we replace the training, development and in-domain test document corpus as Re-DocRED (Tan et al., 2022b), a revised version of DocRED with more complete annotations. Then we follow the same split of relation types as FREDo and sample 15k episodes for in-domain evaluation. The cross-domain test episodes remain the same with FREDo. We also follow Popovic and Färber (2022) to calculate the average class number $N$ and average support instance number per class $K$ across test episodes in ReFREDo, as shown in Table 1. An overview of the relation types and total instance number per relation of two

benchmarks is listed in Appendix A. Following Popovic and Färber (2022), we use macro $F_1$ to evaluate the overall performance.

### 4.2 Baselines

We compare our method with four baselines of FREDo (Popovic and Färber, 2022): **DL-Base** is an initial baseline which uses the pre-trained language model without fine-tuning. **DL-MNAV** is a metric-based approach built upon the state-of-the-art supervised DocRE method (Zhou et al., 2021) and few-shot sentence-level relation extraction method (Sabo et al., 2021). **DL-MNAV**$_{SIE}$ uses all individual support entity pairs during inference instead of averaging their embeddings into a single prototype to improve DL-MNAV for cross-domain tasks. **DL-MNAV**$_{SIE+SBN}$ uses NOTA instances as additional NOTA prototypes during training and only uses NOTA instances during inference to further improve DL-MNAV$_{SIE}$ for cross-domain tasks. Besides, we also evaluate the supervised DocRE model by learning on the whole training corpus and fine-tuning on the support set. Here we choose **KDDocRE** (Tan et al., 2022a) which is the state-of-the-art public-available supervised DocRE method. For a fair comparison, we follow Popovic and Färber (2022) to use BERT-base (Devlin et al., 2019) as the encoder in our approach. We present the implementation details in Appendix B.

### 4.3 Main Results

The main results on FREDo and ReFREDo are shown in Table 2. We have following observations from the experimental results: (1) RAPL achieves significantly better average results on two benchmarks compared to baseline approaches (2.50% $F_1$ on FREDo and 2.72% $F_1$ on ReFREDo), demonstrating the superiority of our method. (2) RAPL consistently outperforms the best baseline method (which varies in different task settings) in each task

| Model / $F_1$ | In-Domain | | Cross-Domain | |
|---|---|---|---|---|
| | 1-Doc | 3-Doc | 1-Doc | 3-Doc |
| RAPL | **15.20** | **16.35** | **3.51** | **5.48** |
| − RCL | 14.13 | 15.32 | 2.51 | 4.63 |
| − IBPC − RCL | 13.36 | 13.96 | 1.68 | 3.10 |
| − IBPC − RCL + SCL | 13.51 | 13.88 | 1.95 | 3.23 |
| − TNPG | 14.50 | 15.69 | 2.99 | 4.72 |

Table 3: Ablation study results on ReFREDo.

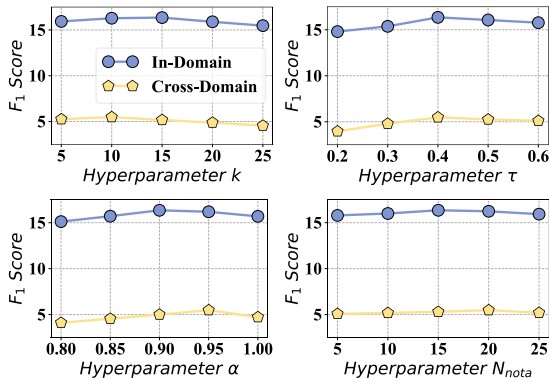

Figure 4: Effect of hyperparameters $k$, $\tau$, $\alpha$ and $N_{nota}$ on RAPL under the 3-Doc task setting in ReFREDo.

setting, making it more versatile than previous approaches. (3) RAPL shows more improvements on in-domain tasks compared to cross-domain tasks. This further reflects the greater challenge posed by cross-domain settings. (4) The performance of RAPL on 3-Doc tasks is consistently higher than that on 1-Doc tasks, which is not always guaranteed for the best baseline method, demonstrating the better scalability of RAPL. (5) The in-domain performance of all methods on ReFREDo is significantly higher than that on FREDo, while this performance gap is not reflected between two benchmarks under the cross-domain setting. This suggests that a higher-quality training set may not effectively resolve the domain adaption problem. (6) The performance of KDDocRE is not satisfactory, indicating that the supervised DocRE method may not adapt well to few-shot scenarios.

## 4.4 Ablation Study

We conduct an ablation study on ReFREDo to investigate the influence of each module in our method. Specifically, for "−RCL", we remove the relation-weighted contrastive learning method; for "−IBPC−RCL", we further remove the instance-based relation prototype construction method, and only obtain the pair-level embedding for each support entity pair in the same way as query entity pairs; for "−IBPC−RCL+SCL", we add a supervised contrastive learning objective (Khosla et al., 2020; Gunel et al., 2021) into the "−IBPC−RCL" model, where we treat those entity pairs sharing common relations as positive pairs, else as negative pairs; for "−TNPG", we remove the task-specific NOTA prototype generation strategy, and directly treat the base NOTA prototypes as final NOTA prototypes. The average results are shown in Table 3. We can observe that the performance of model "−RCL" and "−TNPG" drops to varying degrees compared to RAPL, and the model "−IBPC−RCL" performs even worse than "−RCL", demonstrating the effectiveness of each module in our method. Be-

sides, integrating the contrastive objective at pair-level do not bring significant improvements, which indicates the importance of learning instance-level support embeddings.

## 4.5 Analyses and Discussions

**Effect of Hyperparameters.** We investigate the impact of different hyperparameters on the performance of our approach. We conduct the experiments on 3-Doc tasks in ReFREDo. As shown in Figure 4, we can observe that: (1) For the hyperparameter $k$ which controls the derivation of instance-level attention, the best value for in-domain tasks are larger than cross-domain tasks, which may be related to the longer document in in-domain corpus. (2) An appropriate temperature hyperparameter $\tau$ (around 0.4) in the contrastive objective is crucial for the synergy with classification objective and the overall model performance. (3) Blindly reducing the hyperparameter $\alpha$ to increase the weight of support NOTA instances in NOTA prototypes may have a negative impact on the learning of NOTA prototypes. (4) Compared to other hyperparameters, the model is not very sensitive to the number of NOTA prototypes $N_{nota}$ within a certain range.

**Support Embeddings Visualization.** To intuitively illustrate the advantage of our proposed method, we select three semantically-close relation types from the in-domain 3-Doc test corpus of ReFREDo and sample ten support instances for each relation type, then use t-SNE for visualization (Van der Maaten and Hinton, 2008), as shown in Figure 5. Apart from two model variants in ablation study, we also experiment with RAPL−RCL+SCL, which replaces the relation-weighted contrastive loss with the supervised con-

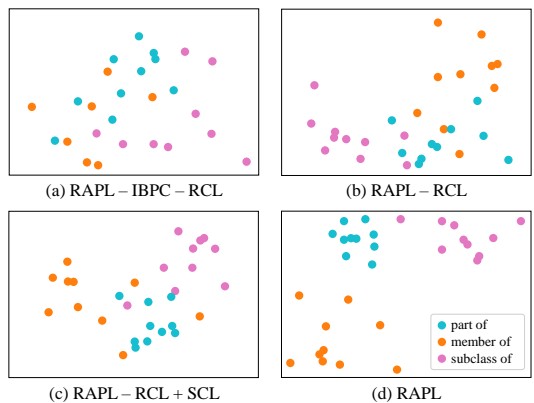

(a) RAPL − IBPC − RCL

(b) RAPL − RCL

(c) RAPL − RCL + SCL

(d) RAPL

Figure 5: Visualization of support entity pair embeddings for RAPL−IBPC−RCL and support relation instance embeddings for RAPL−RCL, RAPL−RCL+SCL and RAPL.

| Model / $F_1$ | NR∈ [0%, 95%) | NR∈ [95%, 97%) | NR∈ [97%, 99%) | NR∈ [99%, 100%] |
|---|---|---|---|---|
| RAPL − TNPG | 23.11 | 18.65 | 17.11 | 5.55 |
| RAPL | 23.51 (↑0.40) | 19.12 (↑0.47) | 17.76 (↑0.65) | 6.66 (↑1.11) |

Table 4: Model performance on episodes with different NOTA rates (abbreviated "NR") in in-domain 3-Doc test set of ReFREDo.

trastive loss (Khosla et al., 2020; Gunel et al., 2021) at instance level. Since some entity pairs express both the "part of" and "member of" relation, or both the "part of" and "subclass of" relation, we only visualize "part of" relation for RAPL−IBPC−RCL in Figure 5(a). We can observe that the support instance embeddings learned by RAPL−RCL improve the support pair embeddings learned by RAPL−IBPC−RCL, demonstrating the effectiveness of instance-level embeddings for relation prototype construction. Besides, although incorporating instance-level supervised contrastive objective forms more compact clusters, the distinction among three relation types is still insufficient. As shown in Figure 5(d), our proposed relation-weighted contrastive learning method better distinguishes the three relation types.

**Performance vs. NOTA Rate of Episodes.** We further explore the impact of task-specific NOTA prototype generation strategy on the performance improvements. We divide the in-domain 3-Doc test episodes of ReFREDo into disjoint subsets according to the NOTA rate of each episode, i.e., the proportion of NOTA entity pairs to the total number of entity pairs in the query document of an episode. We establish four subsets, corresponding to the

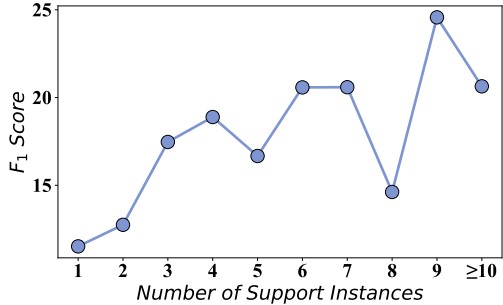

Figure 6: Performance of RAPL under different number of support relation instances on in-domain 3-Doc tasks of ReFREDo.

scenarios where the NOTA rate falls within [0%, 95%), [95%, 97%), [97%, 99%) and [99%, 100%], respectively. Then we evaluate models trained with or without task-specific NOTA prototype generation strategy on each subset. The experiment results are shown in Table 4. It is observed that the task-specific NOTA prototype generation strategy has brought improvement on each subset. More importantly, the performance gain gets larger as the NOTA rate increases. It demonstrates that the task-specific NOTA prototype generation strategy conduces to the capture of NOTA semantics for derived NOTA representations, especially in those episodes with more NOTA query pairs involved.

**Performance vs. Number of Support Relation Instances.** We also analyze the effect of the number of support relation instances on the overall performance. We conduct the experiments on in-domain 3-Doc tasks in ReFREDo benchmark. For each relation type in each test episode, we calculate the number of support instances of that relation type in the episode. Here we divide the number of support instances into 10 categories, where the first 9 categories correspond to 1 to 9, and the last category corresponds to cases where the number of support instances is greater than or equal to 10. Then we evaluate the performance of RAPL method on each of these categories, as shown in Figure 6. We can observe that the performance of RAPL generally exhibits an upward trend as the number of support relation instances increases, while fluctuations also appear at certain points. This indicates that the proposed method demonstrates a certain level of scalability, but the performance may not be perfectly positively correlated with the number of support relation instances.

**Preliminary Exploration of LLM for FSDLRE.**
Recently large language models (LLM) (Brown et al., 2020; Touvron et al., 2023) have achieved promising results in many few-shot tasks through in-context learning (Wei et al., 2022; Rubin et al., 2022). Also some works focus on leveraging LLM to solve few-shot information extraction problems (Ma et al., 2023c; Ye et al., 2023; Wadhwa et al., 2023). However, most studies mainly target sentence-level tasks. Therefore, we conduct a preliminary experiment using gpt-3.5-turbo[3] to explore the performance of LLM on FSDLRE tasks. Due to the input length limit, we only experiment on 1-Doc setting. We randomly select 1000 episodes from the in-domain test episodes of Re-FREDo and design an in-context learning prompt template that includes task description, demonstration and query (detailed in Appendix C). The experimental results show that gpt-3.5-turbo achieves only **12.98%** macro $F_1$, even lower than some baseline methods. Although the test may not fully reflect the capabilities of LLM, we argue that FSDLRE remains a challenging problem even in the era of LLM.

## 5 Related Work

**Sentence-Level Relation Extraction.** Relation extraction is a pivot task of information extraction (Hu et al., 2021, 2023b; Yang et al., 2023). Early studies mainly focus on predicting the relation between two entities within a single sentence. A variety of pattern based (Pantel and Pennacchiotti, 2006; Mintz et al., 2009; Qu et al., 2018) and neural based (Zhang et al., 2018; Baldini Soares et al., 2019; Hu et al., 2020; Liu et al., 2022c) models have achieved satisfactory results. Nevertheless, sentence-level relation extraction has significant limitations in terms of extraction scope and scale. The demand for cross-sentence and large-scale relation extraction has led to a surge of research interest in document-level relation extraction (Quirk and Poon, 2017; Yao et al., 2019).

**Document-Level Relation Extraction.** Most of existing DocRE studies are grounded on a data-driven supervised scenario, and can be generally categorized into graph-based and sequence-based approaches. Graph-based methods (Zeng et al., 2020; Xu et al., 2021b; Zhang et al., 2021; Xu et al., 2022; Duan et al., 2022) typically abstract the document by graph structures and perform inference with graph neural networks. Sequence-based methods (Xu et al., 2021a; Tan et al., 2022a; Yu et al., 2022; Xiao et al., 2022; Ma et al., 2023b) encode the long-distance contextual dependencies with transformer-only architectures. Both categories of methods have achieved impressive results in DocRE. However, the reliance on large-scale annotated documents makes these methods difficult to adapt to low-resource scenarios (Li et al., 2023; Hu et al., 2023a).

**Few-Shot Document-Level Relation Extraction.**
To tackle the data scarcity problem prevalent in real-world DocRE scenarios, Popovic and Färber (2022) formulate DocRE into a few-shot learning task. To accomplish the task, they propose multiple metric-based models built upon the state-of-the-art supervised DocRE method (Zhou et al., 2021) and few-shot sentence-level relation extraction method (Sabo et al., 2021), aiming to address different task settings. We note that for an effective metric-based FSDLRE method, the prototype of each class should accurately capture the corresponding relational semantics. However, this can be challenging for existing methods due to their coarse-grained relation prototype learning strategy and "one-for-all" NOTA prototype learning strategy. In this work, we propose a relation-aware prototype learning method to better capture the relational semantics for prototype representations.

## 6 Conclusion

In this paper, we propose RAPL, a novel relation-aware prototype learning method for FSDLRE. We reframe the construction of relation prototypes into instance level and further propose a relation-weighted contrastive learning method to jointly refine the relation prototypes. Moreover, we design a task-specific NOTA prototype generation strategy to better capture the NOTA semantics in each task. Experiment results and further analyses demonstrate the superiority of our method and effectiveness of each component. For future work, we would like to transfer our method to other few-shot document-level IE tasks such as few-shot document-level event argument extraction, which shares similar task structure with FSDLRE.

## Limitations

Firstly, the incorporation of relation encoder and the search process for support NOTA instances add

---

[3] https://platform.openai.com/docs/models/gpt-3-5

to both memory and time expenses. This motivates us to further refine the overall efficiency of our proposed method. Secondly, the assumption that the entity information should be specified may affect the robustness of the method (Liu et al., 2022b). We have noticed that in supervised scenarios, some recent DocRE studies explore the joint entity and relation extraction to circumvent this assumption (Eberts and Ulges, 2021; Xu and Choi, 2022; Zhang et al., 2023). We believe it is beneficial to investigate end-to-end DocRE in few-shot scenarios, where the RAPL method may shed some lights on future work. Lastly, the performance gain of RAPL on cross-domain tasks is lower than that on in-domain tasks. An intriguing avenue for future research is to explore techniques for better performance on cross-domain tasks, e.g., data augmentation (Hu et al., 2023c) and structured knowledge guidance (Liu et al., 2022a; Ma et al., 2023a).

## Acknowledgements

We sincerely thank the anonymous reviewers for their valuable comments. The work was supported by the National Key Research and Development Program of China (No. 2019YFB1704003), the National Nature Science Foundation of China (No. 62021002), Tsinghua BNRist and Beijing Key Laboratory of Industrial Bigdata System and Application.

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

## A  Relation Types in Benchmarks

In Table 5~9, we list the relation types of training, development, in-domain test and cross-domain test document corpus in FREDo and ReFREDo. We present the name, description and total instance number of each relation type.

## B  Implementation Details

We implement our method with Pytorch (Paszke et al., 2019) and Huggingface's Transformers (Wolf et al., 2020). We use AdamW (Loshchilov and Hutter, 2019) for optimization with a linear warmup for the first 4% steps followed by a linear decay to 0. We train the model for 50k episodes and perform early stopping based on the macro $F_1$ on the development set. We take the learning rate as 1e-5. The episode number per batch during training is set to 4. We clip the gradients to a max norm of 1.0. The hyperparameters $k$, $\tau$, $N_{nota}$, $\alpha$ and $\lambda$ are set to 15, 0.4, 15, 0.9 and 0.1 for in-domain tasks, 10, 0.4, 20, 0.95 and 0.1 for cross-domain tasks. All hyperparameters are tuned on the development set. We report the mean and standard deviation of macro $F_1$ by five training trials with different random seeds. All experiments are conducted with one Tesla V100-32G GPU. For the baseline results on ReFREDo, we reimplement all baseline models with official public codes for comparison.

## C  In-Context Learning Prompt Template for 1-Doc FSDLRE Tasks

**In-context learning prompt template for 1-Doc FSDLRE tasks:**

Given a target relation type list, a document, and all entity mentions of each entity in the document, please identify all valid given relation types between any two given entities in the document.

Target relation type names and descriptions:

<Relation Name 1>: <Relation Description 1>

<Relation Name 2>: <Relation Description 2>

......

Document (each entity mention is enclosed by the ID of the entity):

<Document>

ID and mentions of each entity in the document:

[1]: <Mention 1 of Entity 1>; <Mention 2 of Entity 1>; ......

[2]: <Mention 1 of Entity 2>; <Mention 2 of Entity 2>; ......

......

All non-duplicate valid "subject entity"-"relation type"-"object entity" triples in the document (output format: "entity ID"-"relation type name"-"entity ID", e.g., [1]-country-[2]; one triple per line):

[<Entity ID>]-<Relation Name>-[<Entity ID>]

[<Entity ID>]-<Relation Name>-[<Entity ID>]

......

Document (each entity mention is enclosed by the ID of the entity):

<Document>

ID and mentions of each entity in the document:

[1]: <Mention 1 of Entity 1>; <Mention 2 of Entity 1>; ......

[2]: <Mention 1 of Entity 2>; <Mention 2 of Entity 2>; ......

......

All non-duplicate valid "subject entity"-"relation type"-"object entity" triples in the document (output format: "entity ID"-"relation type name"-"entity ID", e.g., [1]-country-[2]; one triple per line):

## D    Case Study

We select a representative in-domain 1-Doc episode from ReFREDo benchmark for a case study, as shown in Table 10, which intuitively illustrates both the superiority and the bottleneck of the RAPL method. We can observe that: (1) RAPL corrected a false negative prediction of relation P361 for the baseline method. Note that the entity pair of the only instance for P361 in the support document also expresses the relation P140 and P279, and the relation P279 and P463 are semantically close to P361. This suggests the effectiveness of instance-level prototype construction and relation-weighted contrastive refinement in RAPL method. (2) RAPL corrected a false positive prediction of relation P140 between entity *Mayflower* and *Episcopal Diocese of Connecticut*. This pair of entities actually does not convey any target relationship in the query document. Such case may benefit from the task-specific NOTA prototype generation strategy, which better characters the NOTA semantics. (3) When the patterns or reasoning processes of the relation instances in query document differ significantly from the support instances with same relation type, RAPL often struggles with extraction. Also, RAPL tends to exhibit cases of over-prediction, resulting in relatively lower precision. Although the proposed RAPL method achieves certain improvements, the overall performance of few-shot DocRE still lags far behind the supervised setting, and how to overcome the two aforementioned challenges is worth further exploration in future research.

| Wikidata ID | Name | Description | # Instances in FREDo | # Instances in ReFREDo |
|---|---|---|---|---|
| P131 | located in the administrative territorial entity | the item is located on the territory of the following administrative entity | 4193 | 20402 |
| P577 | publication date | date or point in time a work is first published or released | 1142 | 1621 |
| P175 | performer | performer involved in the performance or the recording of a work | 1052 | 1773 |
| P569 | date of birth | date on which the subject was born | 1044 | 1172 |
| P570 | date of death | date on which the subject died | 805 | 1000 |
| P527 | has part | part of this subject. Inverse property of "part of" | 632 | 2313 |
| P161 | cast member | actor performing live for a camera or audience | 621 | 919 |
| P264 | record label | brand and trademark associated with the marketing of subject music recordings and music videos | 583 | 923 |
| P19 | place of birth | most specific known (e.g. city instead of country, or hospital instead of city) birth location of a person, animal or fictional character | 511 | 692 |
| P54 | member of sports team | sports teams or clubs that the subject currently represents or formerly represented | 379 | 379 |
| P40 | child | subject has the object in their family as their offspring son or daughter (independently of their age) | 360 | 703 |
| P30 | continent | continent of which the subject is a part | 356 | 761 |
| P69 | educated at | educational institution attended by the subject | 316 | 503 |
| P400 | platform | platform for which a work has been developed or released / specific platform version of a software developed | 304 | 460 |
| P26 | spouse | the subject has the object as their spouse (husband, wife, partner, etc.) | 303 | 640 |
| P607 | conflict | battles, wars or other military engagements in which the person or item participated | 275 | 575 |
| P22 | father | male parent of the subject | 273 | 466 |
| P159 | headquarters location | specific location where an organization's headquarters is or has been situated | 264 | 263 |
| P178 | developer | organisation or person that developed this item | 238 | 402 |
| P170 | creator | maker of a creative work or other object (where no more specific property exists) | 231 | 410 |
| P1344 | participant of | event a person or an organization was a participant in, inverse of "participant" | 223 | 1168 |
| P6 | head of government | head of the executive power of this town, city, municipality, state, country, or other governmental body | 210 | 368 |
| P127 | owned by | owner of the subject | 208 | 389 |
| P20 | place of death | most specific known (e.g. city instead of country, or hospital instead of city) death location of a person, animal or fictional character | 203 | 281 |
| P108 | employer | person or organization for which the subject works or worked | 196 | 421 |
| P206 | located in or next to body of water | sea, lake or river | 194 | 431 |
| P156 | followed by | immediately following item in some series of which the subject is part | 192 | 506 |
| P710 | participant | person, group of people or organization (object) that actively takes/took part in the event (subject) | 191 | 1168 |
| P155 | follows | immediately prior item in some series of which the subject is part | 188 | 506 |
| P166 | award received | award or recognition received by a person, organisation or creative work | 173 | 340 |
| P276 | location | location of the item, physical object or event is within | 172 | 336 |

Table 5: Relation types of training document corpus in FREDo and ReFREDo (continued on next page).

| Wikidata ID | Name | Description | # Instances in FREDo | # Instances in ReFREDo |
|---|---|---|---|---|
| P123 | publisher | organization or person responsible for publishing books, periodicals, games or software | 172 | 298 |
| P58 | screenwriter | author(s) of the screenplay or script for this work | 156 | 237 |
| P1412 | languages spoken, written or signed | language(s) that a person speaks or writes, including the native language(s) | 155 | 366 |
| P449 | original network | network(s) the radio or television show was originally aired on, including | 152 | 264 |
| P800 | notable work | notable scientific, artistic or literary work, or other work of significance among subject's works | 150 | 3055 |
| P706 | located on terrain feature | located on the specified landform | 137 | 293 |
| P37 | official language | language designated as official by this item | 119 | 281 |
| P162 | producer | producer(s) of this film or music work (film: not executive producers, associate producers, etc.) | 119 | 249 |
| P580 | start time | indicates the time an item begins to exist or a statement starts being valid | 110 | 222 |
| P241 | military branch | branch to which this military unit, award, office, or person belongs | 108 | 191 |
| P937 | work location | location where persons were active | 104 | 204 |
| P31 | instance of | that class of which this subject is a particular example and member. (Subject typically an individual member with Proper Name label.) | 103 | 225 |
| P585 | point in time | time and date something took place, existed or a statement was true | 96 | 191 |
| P403 | mouth of the watercourse | the body of water to which the watercourse drains | 95 | 200 |
| P749 | parent organization | parent organization of an organisation, opposite of subsidiaries | 92 | 230 |
| P36 | capital | primary city of a country, state or other type of administrative territorial entity | 85 | 178 |
| P205 | basin country | country that have drainage to/from or border the body of water | 85 | 174 |
| P172 | ethnic group | subject's ethnicity (consensus is that a VERY high standard of proof is needed for this field to be used. In general this means 1) the subject claims it him/herself, or 2) it is widely agreed on by scholars, or 3) is fictional and portrayed as such). | 79 | 155 |
| P576 | dissolved, abolished or demolished | date or point in time on which an organisation was dissolved/disappeared or a building demolished | 79 | 181 |
| P1376 | capital of | country, state, department, canton or other administrative division of which the municipality is the governmental seat | 76 | 178 |
| P171 | parent taxon | closest parent taxon of the taxon in question | 75 | 117 |
| P740 | location of formation | location where a group or organization was formed | 62 | 102 |
| P840 | narrative location | the narrative of the work is set in this location | 48 | 83 |
| P676 | lyrics by | author of song lyrics | 36 | 79 |
| P551 | residence | the place where the person is, or has been, resident | 35 | 66 |
| P1336 | territory claimed by | administrative divisions that claim control of a given area | 33 | 59 |
| P1365 | replaces | person or item replaced | 18 | 96 |
| P737 | influenced by | this person, idea, etc. is informed by that other person, idea, etc. | 9 | 22 |
| P190 | sister city | twin towns, sister cities, twinned municipalities and other localities that have a partnership or cooperative agreement, either legally or informally acknowledged by their governments | 4 | 8 |
| P1198 | unemployment rate | portion of a workforce population that is not employed | 2 | 2 |
| P807 | separated from | subject was founded or started by separating from identified object | 2 | 8 |

Table 6: Relation types of training document corpus in FREDo and ReFREDo (continued).

| Wikidata ID | Name | Description | # Instances in FREDo | # Instances in ReFREDo |
|---|---|---|---|---|
| P27 | country of citizenship | the object is a country that recognizes the subject as its citizen | 2689 | 4665 |
| P150 | contains administrative territorial entity | (list of) direct subdivisions of an administrative territorial entity | 2004 | 3369 |
| P571 | inception | date or point in time when the organization/subject was founded/created | 475 | 868 |
| P50 | author | main creator(s) of a written work (use on works, not humans) | 320 | 489 |
| P1441 | present in work | work in which this fictional entity or historical person is present | 299 | 669 |
| P57 | director | director(s) of this motion picture, TV-series, stageplay, video game or similar | 246 | 341 |
| P179 | series | subject is part of a series, whose sum constitutes the object | 144 | 245 |
| P136 | genre | a creative work's genre or an artist's field of work | 111 | 239 |
| P112 | founded by | founder or co-founder of this organization, religion or place | 100 | 204 |
| P137 | operator | person or organization that operates the equipment, facility, or service | 95 | 192 |
| P355 | subsidiary | subsidiary of a company or organization, opposite of parent company | 92 | 230 |
| P176 | manufacturer | manufacturer or producer of this product | 83 | 144 |
| P86 | composer | person(s) who wrote the music | 79 | 171 |
| P488 | chairperson | presiding member of an organization, group or body | 63 | 145 |
| P1056 | product or material produced | material or product produced by a government agency, business, industry, facility, or process | 36 | 65 |
| P1366 | replaced by | person or item which replaces another | 36 | 96 |

Table 7: Relation types of development document corpus in FREDo and ReFREDo.

| Wikidata ID | Name | Description | # Instances in FREDo | # Instances in ReFREDo |
|---|---|---|---|---|
| P17 | country | sovereign state of this item; don't use on humans | 2831 | 5505 |
| P495 | country of origin | country of origin of the creative work or subject item | 212 | 455 |
| P361 | part of | object of which the subject is a part. Inverse property of "has part" | 194 | 900 |
| P3373 | sibling | the subject has the object as their sibling (brother, sister, etc.) | 134 | 274 |
| P463 | member of | organization or club to which the subject belongs | 113 | 578 |
| P102 | member of political party | the political party of which this politician is or has been a member | 98 | 98 |
| P1001 | applies to jurisdiction | the item (an institution, law, public office ...) belongs to or has power over or applies to the value (a territorial jurisdiction: a country, state, municipality, ...) | 83 | 485 |
| P140 | religion | religion of a person, organization or religious building, or associated with this subject | 82 | 184 |
| P674 | characters | characters which appear in this item (like plays, operas, operettas, books, comics, films, TV series, video games) | 74 | 204 |
| P194 | legislative body | legislative body governing this entity; political institution with elected representatives, such as a parliament/legislature or council | 56 | 119 |
| P118 | league | league in which team or player plays or has played in | 56 | 126 |
| P35 | head of state | official with the highest formal authority in a country/state | 51 | 131 |
| P272 | production company | company that produced this film, audio or performing arts work | 36 | 79 |
| P279 | subclass of | all instances of these items are instances of those items; this item is a class (subset) of that item | 36 | 86 |
| P364 | original language of work | language in which a film or a performance work was originally created | 30 | 55 |
| P582 | end time | indicates the time an item ceases to exist or a statement stops being valid | 23 | 53 |
| P25 | mother | female parent of the subject | 15 | 59 |
| P39 | position held | subject currently or formerly holds the object position or public office | 8 | 19 |

Table 8: Relation types of in-domain test document corpus in FREDo and ReFREDo.

| SciERC ID | Name | Description | # Instances in FREDo | # Instances in ReFREDo |
|---|---|---|---|---|
| used-for | used for | subject is used for the object; subject models the object; object is trained on the subject; subject exploits the object; object is based on the subject. | 2415 | 2415 |
| conjunction | conjunction | function as similar role or use/incorporate with. | 577 | 577 |
| hyponym-of | hyponym of | subject is a hyponym of the object; subject is a type of the object. | 477 | 477 |
| evaluate-for | evaluate for | evaluate for | 447 | 447 |
| part-of | part of | subject is a part of the object. | 268 | 268 |
| feature-of | feature of | subject belongs to the object; subject is a feature of the object; subject is under the object domain. | 264 | 264 |
| compare | compare | compare two models/methods, or listing two opposing entities. | 232 | 232 |

Table 9: Relation types of cross-domain test document corpus in FREDo and ReFREDo.

**Support Document:**

*Adolfo Nicolás Pachón* (born *29 April 1936*), is a *Spanish* priest of the *Roman Catholic Church.* He was the thirtieth Superior General of the *Society of Jesus*, the largest religious order in the *Roman Catholic Church. Nicolás*, after consulting with Pope *Francis*, determined to resign after his 80th birthday, and initiated the process of calling a *Jesuit General Congregation* to elect his successor. Until the resignation of his predecessor, *Peter Hans Kolvenbach*, it was not the norm for a Jesuit Superior General to resign; they, like the great majority of the Popes up until *Benedict XVI*, generally served until death. However, the *Jesuit* constitutions include provision for a resignation. In *October 2016* the thirty-sixth General Congregation of the *Society of Jesus* appointed his successor, *Arturo Sosa* from *Venezuela.*

**Support Relation Instances:**
**P1001 [applies to jurisdiction]:** *<Arturo Sosa - P1001 - Venezuela>*
**P463 [member of]:** *<Adolfo Nicolás Pachón - P463 - Society of Jesus>*; *<Arturo Sosa - P463 - Society of Jesus>*
**P35 [head of state]:** *<Venezuela - P35 - Arturo Sosa>*
**P279 [subclass of]:** *<Society of Jesus - P279 - Roman Catholic Church>*; *<Jesuit - P279 - Roman Catholic Church>*
**P140 [religion]:** *<Adolfo Nicolás Pachón - P140 - Roman Catholic Church>*; *<Peter Hans Kolvenbach - P140 - Roman Catholic Church>*; *<Benedict XVI - P140 - Roman Catholic Church>*; *<Arturo Sosa - P140 - Roman Catholic Church>*; *<Francis - P140 - Roman Catholic Church>*; *<Society of Jesus - P140 - Roman Catholic Church>*
**P361 [part of]:** *<Society of Jesus - P361 - Roman Catholic Church>*

**Query Document:**

*Chauncey Bunce Brewster* (*September 5, 1848 – April 9, 1941*) was the fifth Bishop of the *Episcopal Diocese of Connecticut. Brewster* was born in *Windham, Connecticut*, to the Reverend *Joseph Brewster* and *Sarah Jane Bunce Brewster.* His father was rector of *St. Paul's Church* in *Windham* and later became rector of *Christ Church* in *New Haven, Connecticut.* His younger brother was the future bishop *Benjamin Brewster.* The family were descendants of *Mayflower* passenger *William Brewster. Brewster* attended *Hopkins Grammar School*, then went to *Yale College*, where he graduated in *1868.* At *Yale* he was elected *Phi Beta Kappa* and was a member of *Skull and Bones.* He attended *Yale's Berkeley Divinity School* the following year. He was consecrated as a bishop on *October 28, 1897.* He was a coadjutor bishop before being diocesan bishop from *1899* to *1928.*

**Gold Outputs for Query Document:**

*<Chauncey Bunce Brewster - P463 - Phi Beta Kappa>*; *<Episcopal Diocese of Connecticut - P1001 - Connecticut>*; *<Berkeley Divinity School - P361 - Yale College>*; *<Chauncey Bunce Brewster - P463 - Skull and Bones>*; *<Chauncey Bunce Brewster - P361 - Skull and Bones>*

**Examples of RAPL Method Correcting Errors in DL-MNAV Method:**

(1) Add the triple *<Berkeley Divinity School - P361 - Yale College>*, which is a false negative case for DL-MNAV method.
(2) Drop the triple *<Mayflower - P140 - Episcopal Diocese of Connecticut>*, which is a false positive case for DL-MNAV method.

**Examples of Errors in RAPL Method:**

(1) False negative prediction of the triple *<Chauncey Bunce Brewster - P463 - Phi Beta Kappa>*.
(2) False positive prediction of the triple *<Benjamin Brewster - P1001 - New Haven>*.

Table 10: Case study of an in-domain 1-Doc episode in ReFREDo. Entity mentions are indicated in italics.