# OpenReview forum: "RAPL: A Relation-Aware Prototype Learning Approach for Few-Shot Document-Level Relation Extraction"
_EMNLP/2023/Conference — EMNLP 2023 Main_

### Official Review · Reviewer_e9xn · 2023-08-03

**Typos Grammar Style And Presentation Improvements:** None
**Soundness:** 4

**Excitement:**

3: Ambivalent: It has merits (e.g., it reports state-of-the-art results, the idea is nice), but there are key weaknesses (e.g., it describes incremental work), and it can significantly benefit from another round of revision. However, I won't object to accepting it if my co-reviewers champion it.

**Missing References:**

None

**Paper Topic And Main Contributions:**

This paper studies the task of Few-shot Document-level Relation Extraction, where two issues are explored:1) how to refine relation prototype representations which are distinct to the confusing ones. 2) how to customize the representations of NOTA types for specific tasks. To tackle the issues above, Relation-Aware Prototype Learning method (RAPL) is proposed. For the first issue, the semantics of relations are used as guidance and the prototype is aggregated from all the supporting instances. Besides, a relation-weighted contrastive learning module works to further refine the prototypes. For the second issue, a set of learnable base NOTA prototypes are derived to produce the task-specific prototypes. Experiments on the public benchmarks in two scenarios demonstrate the effectiveness of the proposed method.

**Questions For The Authors:**

1.The computation process to $\bm{a}_{e_h}$, $\bm{a}_{e_t}$ in Line210 should be specified.

2.$a^{(h,r,t)}$ in Eq.5 is not in bold while $a^{(h,r,t)}$ in Line235 is in bold. Are these two symbols the same, or what are the differences between them?

**Reasons To Accept:**

1.The proposed issues worth investigation for Few-shot Doc RE and the proposed method seems reasonable.

2.The proposed method achieves great performances on the public benchmarks.

3.It is glad to see a Preliminary Exploration of LLM for FSDLRE.

**Reasons To Reject:**

1.Some technical details are not clear enough. Please refer to Question to authors for details.

2.An analysis, which is about how the issue of NOTA is tackled, should be added to support the claim.

**Reproducibility:**

3: Could reproduce the results with some difficulty. The settings of parameters are underspecified or subjectively determined; the training/evaluation data are not widely available.

**Reviewer Confidence:**

3: Pretty sure, but there's a chance I missed something. Although I have a good feel for this area in general, I did not carefully check the paper's details, e.g., the math, experimental design, or novelty.

---

> ### Author Rebuttal · Authors · 2023-08-29
>
> We greatly appreciate the reviewers' recognition of the significance and validity of our work. We are also grateful for the constructive comments provided. We have responded to each suggestion pertaining to areas for enhancement in the manuscript, and will integrate all these revisions into the subsequent version of the manuscript.
>
> > (1) "The computation process to $\boldsymbol{a} _{e _{h}}, \boldsymbol{a} _{e _{t}}$ in Line210 should be specified."
>
> Thanks for the constructive suggestion. We apologize that our explanation regarding the computation process of $\boldsymbol{a} _{e _{h}}$ and $\boldsymbol{a} _{e _{t}}$ in Lines 211-214 is not clear enough. Here we elaborate on it more formally. Following the symbol definitions in the manuscript, given the cross token attention matrix $\boldsymbol{A} \in \mathbb{R}^{N _{t}\times N _{t}}$ output by the document encoder, we first denote the attention from "*" symbol before $e _{h}$'s $i$-th mention $m _{i}^{h}$ as the mention-level attention $\boldsymbol{a} _{m _{i}^{h}} \in \mathbb{R}^{N _{t}}$. Then we average the mention-level attention of each mention of entity $e _{h}$ to obtain the entity-level attention $\boldsymbol{a} _{e _{h}} \in \mathbb{R}^{N _{t}}$: $\boldsymbol{a} _{e _{h}}=\frac{1}{N _{e _{h}}} \sum _{i=1}^{N _{e _{h}}} \boldsymbol{a} _{m _{i}^{h}}$, where $N _{e _{h}}$ is the number of mentions of entity $e _{h}$. The calculation process of $\boldsymbol{a} _{e _{t}}$ is similar.
>
> > (2) "$a _{i}^{(h, r, t)}$ in Eq.5 is not in bold while $\boldsymbol{a}^{(h, r, t)}$ in Line235 is in bold. Are these two symbols the same, or what are the differences between them?"
>
> Thank you for the valuable feedback. These two symbols are not the same. As literally shown, $a _{i}^{(h, r, t)}$ in Eq.5 has an additional subscript "$i$" compared to $\boldsymbol{a}^{(h, r, t)}$ in Line 235. In terms of meaning, the latter is a vector ($\boldsymbol{a}^{(h, r, t)} \in \mathbb{R}^{N _{t}}$), representing the instance-level attention over all tokens in the document; while the former is a scalar ($a _{i}^{(h, r, t)} \in \mathbb{R}$), which is precisely the $i$-th dimension of $\boldsymbol{a}^{(h, r, t)}$ and represents the instance-level attention value on the $i$-th token in the document.
>
> > (3) "An analysis, which is about how the issue of NOTA is tackled, should be added to support the claim."
>
> Thank you for the helpful suggestion. We have further explored the impact of task-specific NOTA prototype generation strategy on the performance improvements. We divide the episodes of in-domain 3-Doc setting in ReFREDo benchmark into disjoint subsets according to the NOTA rate of each episode, i.e., the proportion of NOTA entity pairs to the total number of entity pairs in the query document of an episode. We establish four subsets, corresponding to the scenarios where the NOTA rate falls within [0%, 95%), [95%, 97%), [97%, 99%) and [99%, 100%], respectively. Then we evaluate models trained with or without task-specific NOTA prototype generation strategy on each subset. The experiment results are shown in the table below (where 'NR' refers to NOTA rate and 'TNPG' refers to task-specific NOTA prototype generation strategy). It is observed that the task-specific NOTA prototype generation strategy has brought improvement on each subset. And more importantly, the performance gain gets larger as the NOTA rate increases. It demonstrates that by incorporating task-specific NOTA prototype generation strategy, the derived NOTA representations conduce to capture the NOTA semantics, especially in those episodes with more NOTA query pairs involved.
>
> | Model         | NR $\in$ [0%, 95%) $F _1$ | NR $\in$ [95%, 97%) $F _1$ | NR $\in$ [97%, 99%) $F _1$ | NR $\in$ [99%, 100%] $F _1$ |
> | ------------- | :-----------------------: | :------------------------: | :------------------------: | :-------------------------: |
> | RAPL w/o TNPG |          $23.11$          |          $18.65$           |          $17.11$           |           $5.55$            |
> | RAPL          | $23.51$ ($\uparrow 0.40$) | $19.12$ ($\uparrow 0.47$)  | $17.76$ ($\uparrow 0.65$)  |  $6.66$ ($\uparrow 1.11$)   |

---

### Official Review · Reviewer_rUUw · 2023-08-04

**Soundness:** 4

**Excitement:**

4: Strong: This paper deepens the understanding of some phenomenon or lowers the barriers to an existing research direction.

**Paper Topic And Main Contributions:**

The paper proposes the RAPL model for a few-shot document-level relation extraction task. RAPL model follows a relation-aware prototype learning method for generating a better representation of semantic relations. In contrast to previous approaches, which aggregate the representations of all relevant entity pairs for learning the prototype, RAPL computes instance-level representation for each relation and incorporates contrastive learning to distinguish semantically-close relations better. RAPL also handles the "none-of-the-above" (NOTA) case of relation extraction by constructing task-specific NOTA vectors in contrast to the generalized NOTA vector. RAPL is tested on the FREDo benchmark and performs better than the chosen baseline methods. The authors also propose a new benchmark, ReFREDo, which replaces the DocRED version in FREDo with an improved version previously released by Tan et al., 2022b.

**Questions For The Authors:**

Question A: In line 280, what is the intuition behind defining the contrastive loss based on P_{h,r,t} and A_{h,r,t}? More explanation on what these two sets comprise would be helpful.

Question B: What are the different tasks for the NOTA vector (line 344)?

Question C: Since the main performance gain for RAPL comes from instance-level meta-learning, it would be good to analyze the effect of the number of relation-specific instances on the overall performance. The number of instances vs. performance curve should also help to understand the scalability of RAPL when the document length increases, which directly increases the number of entity pairs.

Question D: As mentioned in line 449, does the higher performance on 3-Doc come from more relation instances? It needs to be clarified how scalability could be the reason for better performance on 3-Doc vs. 1-Doc.

Question E: In line 593, it's mentioned that 1000 episodes were directly selected to run the relation extraction task on the ReFREDo benchmark using GPT-3.5. Getting the RAPL performance on this sample would be suitable for better comparison.

Question F: Although RAPL beats the chosen baselines, presenting an error analysis on the cases where RAPL fails would be insightful, motivating further research in this direction. Especially for the NOTA case, it would be helpful to understand whether task-specific learning is beneficial.

Question G: It's understandable that for comparison on the FREDo benchmark, macro-F1 is used. However, listing the relation-specific precision and recall scores would help understand whether the relation-specific instance-level tuning really helps RAPL's performance or if the gain only comes from handling the easy relations in the benchmark dataset.

**Reasons To Accept:**

The paper is very well-written and easy to follow. The RAPL model proposed in the paper handled an important task of the few-shot document-level relation extraction task. Although the work can seem incremental on the FREDo benchmark, the claims and rationales proposed in the paper are intuitive and justified with experimental results. The methodology is explained in detail, and combining contrastive learning with instance-level aggregation for prototype tuning is interesting.

**Reasons To Reject:**

The RAPL model is incremental work, and the architecture uses the weak BERT encoder, which might cause a performance bottleneck. The method assumes that entity mentions should be specified for a query document, making it less robust. One of the paper's main contributions is to build a task-specific NOTA prototype; however, precise experiments to evaluate NOTA relations are missing.

**Reproducibility:**

4: Could mostly reproduce the results, but there may be some variation because of sample variance or minor variations in their interpretation of the protocol or method.

**Reviewer Confidence:**

4: Quite sure. I tried to check the important points carefully. It's unlikely, though conceivable, that I missed something that should affect my ratings.

---

> ### Author Rebuttal · Authors · 2023-08-29
>
> We sincerely appreciate the reviewers' recognition of the structure and innovation of our work. We are thankful for the detailed feedback provided to help us enhance the quality of our manuscript. We will address each point of feedback and incorporate all these revisions into the subsequent version of our manuscript.
>
> > (1) "The architecture uses the weak BERT encoder, which might cause a performance bottleneck."
>
> Thanks for the constructive comment. As mentioned in Lines 430-432, the reason we use BERT-base as the document encoder is to make a fair comparison with all baseline methods (which also employ BERT-base encoder). In fact, other stronger text encoders can also be considered, such as RoBERTa-large, DeBERTa-large, etc. We have replaced the encoder of three best baseline methods and our RAPL method with RoBERTa-large, and conducted experiments on ReFREDo benchmark (without too much hyperparameter tuning). The results are shown in the table below. It can be observed that the stronger RoBERTa-large encoder brings an average improvement of 1.53% $F _1$ across four methods. Meanwhile, our RAPL method still consistently and significantly outperforms the baselines. We also argue that adopting DeBERTa-large may lead to even greater improvements.
>
> | Model                | In-Domain 1-Doc $F _1$ | In-Domain 3-Doc $F _1$ | Cross-Domain 1-Doc $F _1$ | Cross-Domain 3-Doc $F _1$ |
> | -------------------- | :--------------------: | :--------------------: | :-----------------------: | :-----------------------: |
> | DL-MNAV              |    $14.31 \pm 0.79$    |    $15.20 \pm 0.89$    |      $2.21 \pm 0.40$      |      $3.64 \pm 0.34$      |
> | DL-MNAV$ _{SIE}$     |    $14.57 \pm 0.83$    |    $14.07 \pm 0.61$    |      $2.59 \pm 0.38$      |      $4.56 \pm 0.36$      |
> | DL-MNAV$ _{SIE+SBN}$ |    $5.89 \pm 0.22$     |    $7.17 \pm 0.43$     |      $3.82 \pm 0.48$      |      $5.44 \pm 0.18$      |
> | RAPL (Ours)          |    $16.76 \pm 0.86$    |    $18.30 \pm 0.93$    |      $4.55 \pm 0.65$      |      $7.12 \pm 0.62$      |
>
> > (2) "The method assumes that entity mentions should be specified for a query document, making it less robust."
>
> Thank you for pointing out this issue. On one hand, this assumption tightly follows the task definition of FSDLRE as outlined in [1], where it is mentioned that the entity information is provided as part of the input. On the other hand, this assumption could indeed impact the robustness of the method. We have noticed that in supervised scenarios, some recent DocRE studies have started to exploring the joint entity and relation extraction to circumvent this assumption, and have made some progress \[2]\[3]\[4]. We would like to investigate end-to-end DocRE in few-shot scenarios in our future work, where the RAPL method may shed some lights on the design of end-to-end models.
>
> > (3) "One of the paper's main contributions is to build a task-specific NOTA prototype; however, precise experiments to evaluate NOTA relations are missing."
>
> Thank you for the valuable feedback. We have further explored the impact of task-specific NOTA prototype generation strategy on the performance improvements. We divide the episodes of in-domain 3-Doc setting in ReFREDo benchmark into disjoint subsets according to the NOTA rate of each episode, i.e., the proportion of NOTA entity pairs to the total number of entity pairs in the query document of an episode. We establish four subsets, corresponding to the scenarios where the NOTA rate falls within [0%, 95%), [95%, 97%), [97%, 99%) and [99%, 100%], respectively. Then we evaluate models trained with or without task-specific NOTA prototype generation strategy on each subset. The experiment results are shown in the table below (where 'NR' refers to NOTA rate and 'TNPG' refers to task-specific NOTA prototype generation strategy). It is observed that the task-specific NOTA prototype generation strategy has brought improvement on each subset. And more importantly, the performance gain gets larger as the NOTA rate increases. It demonstrates that by incorporating task-specific NOTA prototype generation strategy, the derived NOTA representations conduce to capture the NOTA semantics, especially in those episodes with more NOTA query pairs involved.
>
> | Model         | NR $\in$ [0%, 95%) $F _1$ | NR $\in$ [95%, 97%) $F _1$ | NR $\in$ [97%, 99%) $F _1$ | NR $\in$ [99%, 100%] $F _1$ |
> | ------------- | :-----------------------: | :------------------------: | :------------------------: | :-------------------------: |
> | RAPL w/o TNPG |          $23.11$          |          $18.65$           |          $17.11$           |           $5.55$            |
> | RAPL          | $23.51$ ($\uparrow 0.40$) | $19.12$ ($\uparrow 0.47$)  | $17.76$ ($\uparrow 0.65$)  |  $6.66$ ($\uparrow 1.11$)   |
>
> > (4) "Question A: In line 280, what is the intuition behind defining the contrastive loss based on P_{h,r,t} and A_{h,r,t}? More explanation on what these two sets comprise would be helpful."
>
> Thank you for the constructive suggestion. We apologize for not providing a detailed explanation on these two sets. Following the symbol definitions in the manuscript, for a relation instance $(e _{h}, r, e _{t})$ in the support set, the set $\mathcal{P} _{h,r,t}=\mathcal{S} _{r}\setminus \{(e _{h}, r, e _{t})\} $ contains all other instances in the support set that also express the relation $r$, and the set $\mathcal{A} _{h,r,t}=\mathcal{S}\setminus \{(e _{h}, r, e _{t})\} $ simply contains all other instances in the support set. Therefore, considering $(e _{h}, r, e _{t})$ as an anchor in the contrastive objective, each instance in $\mathcal{P} _{h,r,t}$ constitutes a positive example of $(e _{h}, r, e _{t})$, and the contrastive objective aims to pull close the instances of same relation (i.e. positive examples) and push apart the instances of different relations (i.e. negative examples).
>
> > (5) "Question B: What are the different tasks for the NOTA vector (line 344)?"
>
> We speculate that this comment pertains to the sentence in Lines 306-311, where we mentioned that "the NOTA semantics differs in tasks with different target relation types". Here what we mean is, for two tasks (i.e. episodes) with different target relation types, suppose being R1, R2, R3 and R4, R5, R6 respectively, the "NOTA" of these two tasks can be considered as "none of R1, R2, R3" and "none of R4, R5, R6", which are also different in terms of relational semantics.
>
> > (6) "Question C: Since the main performance gain for RAPL comes from instance-level meta-learning, it would be good to analyze the effect of the number of relation-specific instances on the overall performance. The number of instances vs. performance curve should also help to understand the scalability of RAPL when the document length increases, which directly increases the number of entity pairs."
>
> Thank you for the insightful suggestion. We have further analyzed the effect of the number of support relation instances on the overall performance. We conduct the experiments on in-domain 3-Doc tasks in ReFREDo benchmark. For each relation type in each test episode, we calculate the number of support instances of that relation type in the episode. Here we divide the number of support instances into 10 categories, where the first 9 categories correspond to 1 to 9, and the last category corresponds to cases where the number of support instances is greater than or equal to 10. Then we evaluate the performance of RAPL method on each of these categories. The results are shown in the table below (we denote the number of support relation instances as $K$ in the table; and we will present the relevant findings in a more visual manner using line charts in the subsequent version of our manuscript). We can observe that the performance of RAPL method generally exhibits an upward trend as the number of support relation instances increases, while fluctuations also appear at certain points. This indicates that the RAPL method demonstrates a certain level of scalability, but the performance may not be perfectly positively correlated with the number of support relation instances.
>
> | K      |    1    |    2    |    3    |    4    |    5    |    6    |    7    |    8    |    9    | $\ge$ 10 |
> | ------ | :-----: | :-----: | :-----: | :-----: | :-----: | :-----: | :-----: | :-----: | :-----: | :------: |
> | $F _1$ | $11.52$ | $12.75$ | $17.47$ | $18.89$ | $16.67$ | $20.58$ | $20.59$ | $14.62$ | $24.57$ | $20.64$  |
>
> > (7) "Question D: As mentioned in line 449, does the higher performance on 3-Doc come from more relation instances? It needs to be clarified how scalability could be the reason for better performance on 3-Doc vs. 1-Doc."
>
> Thanks for the suggestion. As mentioned in [1], the logical relationship here is that, the performance variation on 3-Doc setting (which has a higher average number of support instances for each relation type than 1-Doc setting, as shown in Table 1 in the manuscript) vs. 1-Doc setting can be used to measure the scalability of the model. Based on the experimental results we can observe that the performance of RAPL on 3-Doc tasks is consistently higher than that on 1-Doc tasks, which is not completely achieved by some strong baseline methods, demonstrating the better scalability of our proposed method.
>
> > (8) "Question E: In line 593, it's mentioned that 1000 episodes were directly selected to run the relation extraction task on the ReFREDo benchmark using GPT-3.5. Getting the RAPL performance on this sample would be suitable for better comparison."
>
> Thank you for the constructive feedback. It is indeed fairer to compare the results of baseline&RAPL methods on these 1000 sampled episodes with the reported result of gpt-3.5-turbo in the manuscript. We have tested our RAPL method and two strong baseline methods (DL-MNAV, DL-MNAV$ _{SIE}$) on these 1000 sampled episodes, and the results are shown in the table below. Compared to the results obtained on the entire test set of 15k episodes, the performances on sampled episodes do not exhibit significant differences. As a result, similar to the discussion in the manuscript, utilizing a generic LLM for in-context learning on FSDLRE does not demonstrate superior performance compared to the baseline and our RAPL methods.
>
> | Model            | ReFREDo In-Domain 1-Doc $F _1$ |
> | ---------------- | :----------------------------: |
> | DL-MNAV          |        $12.89 \pm 0.93$        |
> | DL-MNAV$ _{SIE}$ |        $13.11 \pm 0.87$        |
> | RAPL (Ours)      |        $15.40 \pm 0.80$        |
> | gpt-3.5-turbo    |            $12.98$             |
>
> > (9) "Question F: Although RAPL beats the chosen baselines, presenting an error analysis on the cases where RAPL fails would be insightful, motivating further research in this direction. Especially for the NOTA case, it would be helpful to understand whether task-specific learning is beneficial."
>
> Thank you for the helpful suggestion. Here we select a representative in-domain 1-Doc episode from ReFREDo benchmark for a case study, as shown below, which intuitively illustrates both the superiority and the bottleneck of the RAPL method (we will also present more cases in the subsequent version of our manuscript). We can observe that: (1) RAPL corrected a false negative prediction of relation P361 for the baseline method. Note that the entity pair of the only instance for P361 in the support document also expresses the relation P140 and P279, and the relation P279 and P463 are semantically close to P361. Thus, this suggests the effectiveness of instance-level prototype construction and relation-weighted contrastive refinement in RAPL method. (2) RAPL corrected a false positive prediction of relation P140 between entity *Mayflower* and *Episcopal Diocese of Connecticut*. This pair of entities actually does not convey any target relationship in the document. There are also several similar false positive corrections, which are not listed here. All of these cases may benefit from the task-specific NOTA prototype generation strategy, which better characters the NOTA semantics. (3) When the patterns or reasoning processes of the relation instances in query document differ significantly from the support instances with same relation type, RAPL often struggles with extraction. Also, RAPL tends to exhibit cases of over-prediction, resulting in relatively lower precision. Although the proposed RAPL method achieves certain progress in FSDLRE, the overall performance of few-shot DocRE still lags far behind the supervised setting, and how to overcome the two aforementioned challenges is worth further exploration in future research.
>
> **Support Document:**
>
> *Adolfo Nicolás Pachón (born 29 April 1936), is a Spanish priest of the Roman Catholic Church. He was the thirtieth Superior General of the Society of Jesus, the largest religious order in the Roman Catholic Church. Nicolás, after consulting with Pope Francis, determined to resign after his 80th birthday, and initiated the process of calling a Jesuit General Congregation to elect his successor. Until the resignation of his predecessor, Peter Hans Kolvenbach, it was not the norm for a Jesuit Superior General to resign; they, like the great majority of the Popes up until Benedict XVI, generally served until death. However, the Jesuit constitutions include provision for a resignation. In October 2016 the thirty-sixth General Congregation of the Society of Jesus appointed his successor, Arturo Sosa from Venezuela.*
>
> **Support Relation Instances:**
>
> *P1001 [applies to jurisdiction]: <Arturo Sosa - P1001 - Venezuela>*
>
> *P463 [member of]: <Adolfo Nicolás Pachón - P463 - Society of Jesus>; <Arturo Sosa - P463 - Society of Jesus>*
>
> *P35 [head of state]: <Venezuela - P35 - Arturo Sosa>*
>
> *P279 [subclass of]: <Society of Jesus - P279 - Roman Catholic Church>; <Jesuit - P279 - Roman Catholic Church>*
>
> *P140 [religion]: <Adolfo Nicolás Pachón - P140 - Roman Catholic Church>; <Peter Hans Kolvenbach - P140 - Roman Catholic Church>; <Benedict XVI - P140 - Roman Catholic Church>; <Arturo Sosa - P140 - Roman Catholic Church>; <Francis - P140 - Roman Catholic Church>; <Society of Jesus - P140 - Roman Catholic Church>*
>
> *P361 [part of]: <Society of Jesus - P361 - Roman Catholic Church>*
>
> **Query Document:**
>
> *Chauncey Bunce Brewster (September 5, 1848 – April 9, 1941) was the fifth Bishop of the Episcopal Diocese of Connecticut. Brewster was born in Windham, Connecticut, to the Reverend Joseph Brewster and Sarah Jane Bunce Brewster. His father was rector of St. Paul's Church in Windham and later became rector of Christ Church in New Haven, Connecticut. His younger brother was the future bishop Benjamin Brewster. The family were descendants of Mayflower passenger William Brewster. Brewster attended Hopkins Grammar School, then went to Yale College, where he graduated in 1868. At Yale he was elected Phi Beta Kappa and was a member of Skull and Bones. He attended Yale's Berkeley Divinity School the following year. He was consecrated as a bishop on October 28, 1897. He was a coadjutor bishop before being diocesan bishop from 1899 to 1928.*
>
> **Gold Outputs for Query Document:**
>
> *<Chauncey Bunce Brewster - P463 - Phi Beta Kappa>; <Episcopal Diocese of Connecticut - P1001 - Connecticut>; <Berkeley Divinity School - P361 - Yale College>; <Chauncey Bunce Brewster - P463 - Skull and Bones>; <Chauncey Bunce Brewster - P361 - Skull and Bones>*
>
> **Examples of RAPL method correcting errors in DL-MNAV method:**
>
> 1. Add the triple <Berkeley Divinity School - P361 - Yale College>, which is a false negative case for DL-MNAV method.
> 2. Drop the triple <Mayflower - P140 - Episcopal Diocese of Connecticut>, which is a false positive case for DL-MNAV method.
>
> **Examples of errors in our RAPL method:**
>
> 1. False negative prediction of the triple <Chauncey Bunce Brewster - P463 - Phi Beta Kappa>.
> 2. False positive prediction of the triple <Benjamin Brewster - P1001 - New Haven>.
>
> > (10) "Question G: It's understandable that for comparison on the FREDo benchmark, macro-F1 is used. However, listing the relation-specific precision and recall scores would help understand whether the relation-specific instance-level tuning really helps RAPL's performance or if the gain only comes from handling the easy relations in the benchmark dataset."
>
> Thanks for the insightful suggestion. To investigate the performance gain of RAPL at a finer granularity of per-relation level, we first compare the per-relation performance of RAPL with the best baseline method on each task setting of ReFREDo benchmark. The results show that, for in-domain setting (with 18 relation types in the test set), RAPL outperforms the baseline method on 17 (1-Doc) and 16 (3-Doc) relation types respectively. For cross-domain setting (with 7 relation types in the test set), RAPL outperforms the baseline method on 5 (1-Doc) an 7 (3-Doc) relation types respectively. This demonstrates that RAPL has improved the predictions for the majority of relation types. Furthermore, in the table below, we have listed the 5 relation types with the highest (the first 5 lines) and lowest (the last 5 lines) $F _1$ scores for the best baseline method (DL-MNAV) on in-domain 3-Doc setting. We present the precision, recall and $F _1$ scores of DL-MNAV and RAPL on these 10 relation types, and compare the variations in $F _1$ scores in the last column. We can observe that, the performance gain of RAPL is similar across the 5 easiest and 5 most difficult relation types of the baseline method. This further suggests that RAPL is capable of handling both easy and hard relation types simultaneously.
>
> | Relation                  |   DL-MNAV $P$    |     RAPL $P$     |   DL-MNAV $R$    |     RAPL $R$     |  DL-MNAV $F _1$  |   RAPL $F _1$    |   $\Delta F _1$   |
> | ------------------------- | :--------------: | :--------------: | :--------------: | :--------------: | :--------------: | :--------------: | :---------------: |
> | sibling                   | $21.30 \pm 1.52$ | $24.94 \pm 2.07$ | $49.27 \pm 8.78$ | $51.07 \pm 5.71$ | $29.60 \pm 2.65$ | $33.38 \pm 2.96$ |  $\uparrow 3.78$  |
> | country of origin         | $15.28 \pm 2.13$ | $13.13 \pm 3.05$ | $58.72 \pm 5.18$ | $50.01 \pm 4.68$ | $24.13 \pm 2.47$ | $20.67 \pm 3.39$ | $\downarrow 3.46$ |
> | league                    | $14.36 \pm 2.38$ | $24.94 \pm 4.99$ | $75.74 \pm 3.67$ | $71.49 \pm 7.07$ | $24.05 \pm 3.35$ | $33.46 \pm 5.98$ |  $\uparrow 9.41$  |
> | country                   | $59.18 \pm 3.54$ | $52.93 \pm 2.43$ | $12.83 \pm 2.32$ | $17.67 \pm 2.80$ | $20.97 \pm 3.20$ | $26.66 \pm 2.59$ |  $\uparrow 5.69$  |
> | production company        | $8.67 \pm 0.61$  | $10.77 \pm 1.51$ | $54.73 \pm 6.30$ | $42.52 \pm 3.61$ | $14.95 \pm 0.97$ | $17.24 \pm 1.83$ |  $\uparrow 2.29$  |
> | religion                  | $4.63 \pm 0.92$  | $11.18 \pm 1.91$ | $11.56 \pm 1.31$ | $11.71 \pm 1.25$ | $6.60 \pm 1.15$  | $11.55 \pm 1.35$ |  $\uparrow 4.95$  |
> | part of                   | $6.36 \pm 0.17$  | $7.99 \pm 0.63$  | $5.12 \pm 0.42$  | $11.69 \pm 0.88$ | $5.66 \pm 0.25$  | $9.45 \pm 0.70$  |  $\uparrow 3.79$  |
> | member of political party | $3.00 \pm 0.38$  | $8.40 \pm 1.50$  | $26.64 \pm 3.30$ | $38.43 \pm 4.09$ | $5.39 \pm 0.67$  | $13.71 \pm 1.84$ |  $\uparrow 8.32$  |
> | subclass of               | $2.18 \pm 0.62$  | $4.72 \pm 0.71$  | $5.40 \pm 2.04$  | $12.39 \pm 3.05$ | $3.08 \pm 0.94$  | $6.79 \pm 1.04$  |  $\uparrow 3.71$  |
> | position held             | $0.65 \pm 0.11$  | $1.98 \pm 1.01$  | $37.55 \pm 2.74$ | $30.20 \pm 1.92$ | $1.29 \pm 0.21$  | $3.69 \pm 1.12$  |  $\uparrow 2.40$  |
>
> **References**
>
> [1] Popovic, N., & Färber, M. (2022, July). Few-Shot Document-Level Relation Extraction. In Proceedings of the 2022 Conference of the North American Chapter of the Association for Computational Linguistics: Human Language Technologies (pp. 5733-5746).
>
> [2] Eberts, M., & Ulges, A. (2021, April). An End-to-end Model for Entity-level Relation Extraction using Multi-instance Learning. In Proceedings of the 16th Conference of the European Chapter of the Association for Computational Linguistics: Main Volume (pp. 3650-3660).
>
> [3] Xu, L., & Choi, J. D. (2022, July). Modeling Task Interactions in Document-Level Joint Entity and Relation Extraction. In Proceedings of the 2022 Conference of the North American Chapter of the Association for Computational Linguistics: Human Language Technologies (pp. 5409-5416).
>
> [4] Zhang, R., Li, Y., & Zou, L. (2023, July). A Novel Table-to-Graph Generation Approach for Document-Level Joint Entity and Relation Extraction. In Proceedings of the 61st Annual Meeting of the Association for Computational Linguistics (Volume 1: Long Papers) (pp. 10853-10865).

---

### Official Review · Reviewer_cDEF · 2023-08-07

**Soundness:** 4

**Excitement:**

4: Strong: This paper deepens the understanding of some phenomenon or lowers the barriers to an existing research direction.

**Missing References:**

Just list a few more recent DocRE paper for citation

1. Evidence-aware Document-level Relation Extraction. CIKM '22
2. EIDER: Evidence-enhanced Document-level Relation Extraction. ACL'22
3. A Novel Table-to-Graph Generation Approach for Document-Level Joint Entity and Relation Extraction. ACL'23

**Paper Topic And Main Contributions:**

The paper studies the Few-shot document-level relation extraction (FSDLRE) problem where we need to identify semantic relations among entities in a document when there are only a few labeled samples available. The authors show the limitations of existing metric-based meta-learning approach and to address the problems, this paper introduces a relation-aware prototype learning method for FSDLRE. This method refines the relation prototypes and creates task-specific NOTA prototypes by utilizing relation descriptions and real NOTA instances. Through extensive experiments, this proposed method has shown to surpass current top-performing methods by an average 2.61% F1 across multiple benchmark settings for FSDLRE.

**Reasons To Accept:**

1. The paper studies an important task and proposes an interesting approach to address the FSDLRE problem.
2. The experiments are overall clear and well structured.
3. The paper is clearly written and easy to access.

**Reasons To Reject:**

Overall I feel the paper is solid, presenting some interesting findings and demonstrate the effectiveness of its proposed approach via multiple experiments. There are a few places where the authors can further improve the current writing:
1. Considering adding one more dataset (maybe modified one existing DocRE benchmark) to demonstrate this proposed method can generalize.
2. Adding a few more case studies in the experiment section to intuitively show the output and also to promote the FSDLRE problem setting.
3. Adding more recent DocRE literature in the reference.

**Reproducibility:**

3: Could reproduce the results with some difficulty. The settings of parameters are underspecified or subjectively determined; the training/evaluation data are not widely available.

**Reviewer Confidence:**

2: Willing to defend my evaluation, but it is fairly likely that I missed some details, didn't understand some central points, or can't be sure about the novelty of the work.

---

> ### Author Rebuttal · Authors · 2023-08-29
>
> We deeply appreciate the reviewers' recognition of the novelty and thoroughness of our work. We are also thankful for the insightful comments provided. We have responded to each suggestion regarding the areas for improvement in the manuscript, and will incorporate all these adjustments into the next version of the manuscript.
>
> > (1) "Considering adding one more dataset (maybe modified one existing DocRE benchmark) to demonstrate this proposed method can generalize."
>
> Thank you for the valuable feedback. We conducted a preliminary experiment on the DWIE dataset [1], which is an entity-centric dataset for document-level multi-task information extraction. DWIE consists of 802 general news articles in English, selected randomly from a corpus collected from Deutsche Welle between 2002 and 2018 [1]. We focus on relation extraction task and follow the preprocessing steps as [2] to obtain a version that meets the required format. We use the whole dataset of DWIE as a novel cross-domain document corpus and also sample 3k episodes as sciERC in FREDo and ReFREDo. Here we randomly select 15 out of 65 relation types in DWIE, where we also remove the relation types annotated in the in-domain training corpus from (Re-)DocRED, like 'spouse of' and 'child of', in order to prevent data leakage between train and test sets. We conducted experiments on ReFREDo benchmark (without too much hyperparameter tuning) and the results are shown in the table below. It is observed that the RAPL method consistently and significantly outperforms three baseline methods, demonstrating better generalization ability. We also notice that the overall results of these methods are higher than that on sciERC, which could be attributed to a smaller domain shift between DWIE and DocRED (news vs. wikipedia) compared to sciERC (science). We will further extend DWIE to training corpus (like DocRED in FREDo) and utilize the entire relation type set, in order to enrich the relevant benchmark dataset.
>
> | Model                | Cross-Domain 1-Doc $F _1$ | Cross-Domain 3-Doc $F _1$ |
> | -------------------- | :-----------------------: | :-----------------------: |
> | DL-MNAV              |      $2.51 \pm 0.40$      |      $3.83 \pm 0.48$      |
> | DL-MNAV$ _{SIE}$     |      $2.86 \pm 0.68$      |      $4.53 \pm 0.30$      |
> | DL-MNAV$ _{SIE+SBN}$ |      $3.79 \pm 0.56$      |      $4.65 \pm 0.73$      |
> | RAPL (Ours)          |      $4.98 \pm 0.60$      |      $6.62 \pm 0.83$      |
>
> > (2) "Adding a few more case studies in the experiment section to intuitively show the output and also to promote the FSDLRE problem setting."
>
> Thank you for the insightful suggestion. Here we select a representative in-domain 1-Doc episode from ReFREDo benchmark for a case study, as shown below, which intuitively illustrates both the superiority and the bottleneck of the RAPL method (we will also present more cases in the subsequent version of our manuscript). We can observe that: (1) RAPL corrected a false negative prediction of relation P361 for the baseline method. Note that the entity pair of the only instance for P361 in the support document also expresses the relation P140 and P279, and the relation P279 and P463 are semantically close to P361. Thus, this suggests the effectiveness of instance-level prototype construction and relation-weighted contrastive refinement in RAPL method. (2) RAPL corrected a false positive prediction of relation P140 between entity *Mayflower* and *Episcopal Diocese of Connecticut*. This pair of entities actually does not convey any target relationship in the document. There are also several similar false positive corrections, which are not listed here. All of these cases may benefit from the task-specific NOTA prototype generation strategy, which better characters the NOTA semantics. (3) When the patterns or reasoning processes of the relation instances in query document differ significantly from the support instances with same relation type, RAPL often struggles with extraction. Also, RAPL tends to exhibit cases of over-prediction, resulting in relatively lower precision. Although the proposed RAPL method achieves certain progress in FSDLRE, the overall performance of few-shot DocRE still lags far behind the supervised setting, and how to overcome the two aforementioned challenges is worth further exploration in future research.
>
> **Support Document:**
>
> *Adolfo Nicolás Pachón (born 29 April 1936), is a Spanish priest of the Roman Catholic Church. He was the thirtieth Superior General of the Society of Jesus, the largest religious order in the Roman Catholic Church. Nicolás, after consulting with Pope Francis, determined to resign after his 80th birthday, and initiated the process of calling a Jesuit General Congregation to elect his successor. Until the resignation of his predecessor, Peter Hans Kolvenbach, it was not the norm for a Jesuit Superior General to resign; they, like the great majority of the Popes up until Benedict XVI, generally served until death. However, the Jesuit constitutions include provision for a resignation. In October 2016 the thirty-sixth General Congregation of the Society of Jesus appointed his successor, Arturo Sosa from Venezuela.*
>
> **Support Relation Instances:**
>
> *P1001 [applies to jurisdiction]: <Arturo Sosa - P1001 - Venezuela>*
>
> *P463 [member of]: <Adolfo Nicolás Pachón - P463 - Society of Jesus>; <Arturo Sosa - P463 - Society of Jesus>*
>
> *P35 [head of state]: <Venezuela - P35 - Arturo Sosa>*
>
> *P279 [subclass of]: <Society of Jesus - P279 - Roman Catholic Church>; <Jesuit - P279 - Roman Catholic Church>*
>
> *P140 [religion]: <Adolfo Nicolás Pachón - P140 - Roman Catholic Church>; <Peter Hans Kolvenbach - P140 - Roman Catholic Church>; <Benedict XVI - P140 - Roman Catholic Church>; <Arturo Sosa - P140 - Roman Catholic Church>; <Francis - P140 - Roman Catholic Church>; <Society of Jesus - P140 - Roman Catholic Church>*
>
> *P361 [part of]: <Society of Jesus - P361 - Roman Catholic Church>*
>
> **Query Document:**
>
> *Chauncey Bunce Brewster (September 5, 1848 – April 9, 1941) was the fifth Bishop of the Episcopal Diocese of Connecticut. Brewster was born in Windham, Connecticut, to the Reverend Joseph Brewster and Sarah Jane Bunce Brewster. His father was rector of St. Paul's Church in Windham and later became rector of Christ Church in New Haven, Connecticut. His younger brother was the future bishop Benjamin Brewster. The family were descendants of Mayflower passenger William Brewster. Brewster attended Hopkins Grammar School, then went to Yale College, where he graduated in 1868. At Yale he was elected Phi Beta Kappa and was a member of Skull and Bones. He attended Yale's Berkeley Divinity School the following year. He was consecrated as a bishop on October 28, 1897. He was a coadjutor bishop before being diocesan bishop from 1899 to 1928.*
>
> **Gold Outputs for Query Document:**
>
> *<Chauncey Bunce Brewster - P463 - Phi Beta Kappa>; <Episcopal Diocese of Connecticut - P1001 - Connecticut>; <Berkeley Divinity School - P361 - Yale College>; <Chauncey Bunce Brewster - P463 - Skull and Bones>; <Chauncey Bunce Brewster - P361 - Skull and Bones>*
>
> **Examples of RAPL method correcting errors in DL-MNAV method:**
>
> 1. Add the triple <Berkeley Divinity School - P361 - Yale College>, which is a false negative case for DL-MNAV method.
> 2. Drop the triple <Mayflower - P140 - Episcopal Diocese of Connecticut>, which is a false positive case for DL-MNAV method.
>
> **Examples of errors in our RAPL method:**
>
> 1. False negative prediction of the triple <Chauncey Bunce Brewster - P463 - Phi Beta Kappa>.
> 2. False positive prediction of the triple <Benjamin Brewster - P1001 - New Haven>.
>
> > (3) "Adding more recent DocRE literature in the reference."
>
> Thanks for the constructive suggestion. We will add references to more recent DocRE literatures, including but not limited to \[3]\[4]\[5]\[6]\[7].
>
> **References**
>
> [1] Zaporojets, K., Deleu, J., Develder, C., & Demeester, T. (2021). DWIE: An entity-centric dataset for multi-task document-level information extraction. Information Processing & Management, 58(4), 102563.
>
> [2] Ru, D., Sun, C., Feng, J., Qiu, L., Zhou, H., Zhang, W., ... & Li, L. (2021, November). Learning Logic Rules for Document-Level Relation Extraction. In Proceedings of the 2021 Conference on Empirical Methods in Natural Language Processing (pp. 1239-1250).
>
> [3] Xu, T., Hua, W., Qu, J., Li, Z., Xu, J., Liu, A., & Zhao, L. (2022, October). Evidence-aware Document-level Relation Extraction. In Proceedings of the 31st ACM International Conference on Information & Knowledge Management (pp. 2311-2320).
>
> [4] Xie, Y., Shen, J., Li, S., Mao, Y., & Han, J. (2022, May). Eider: Empowering Document-level Relation Extraction with Efficient Evidence Extraction and Inference-stage Fusion. In Findings of the Association for Computational Linguistics: ACL 2022 (pp. 257-268).
>
> [5] Xiao, Y., Zhang, Z., Mao, Y., Yang, C., & Han, J. (2022, July). SAIS: Supervising and Augmenting Intermediate Steps for Document-Level Relation Extraction. In Proceedings of the 2022 Conference of the North American Chapter of the Association for Computational Linguistics: Human Language Technologies (pp. 2395-2409).
>
> [6] Zhang, R., Li, Y., & Zou, L. (2023, July). A Novel Table-to-Graph Generation Approach for Document-Level Joint Entity and Relation Extraction. In Proceedings of the 61st Annual Meeting of the Association for Computational Linguistics (Volume 1: Long Papers) (pp. 10853-10865).
>
> [7] Sun, Q., Huang, K., Yang, X., Hong, P., Zhang, K., & Poria, S. (2023). Uncertainty Guided Label Denoising for Document-level Distant Relation Extraction. arXiv preprint arXiv:2305.11029.

---

### Meta-Review · Area_Chair_R1Rh · 2023-09-14

**Recommendation:** 4

**Metareview:**

This paper is well-written and proposes an important research question. The experiment results show that their method is able to solve this problem.

---

### Decision · Program_Chairs · 2023-10-07

**Decision:**

Accept-Main

**Comment:**

This paper is well-written and proposes an important research question. The experiment results show that their method is able to solve this problem.